# Federated Sampling with Langevin Algorithm under Isoperimetry

**Lukang Sun**                                                     *lukang.sun@kaust.edu.sa*
*King Abdullah University of Science and Technology*

**Adil Salim**                                                     *adilsalim@microsoft.com*
*Microsoft Research*

**Peter Richtárik**                                                *peter.richtarik@kaust.edu.sa*
*King Abdullah University of Science and Technology*

**Reviewed on OpenReview:** *https://openreview.net/forum?id=Sj7bFPeR6W*

## Abstract

Federated learning uses a set of techniques to efficiently distribute the training of a machine learning algorithm across several devices, who own the training data. These techniques critically rely on reducing the communication cost—the main bottleneck—between the devices and a central server. Federated learning algorithms usually take an optimization approach: they are algorithms for minimizing the training loss subject to communication (and other) constraints. In this work, we instead take a Bayesian approach for the training task, and propose a communication-efficient variant of the Langevin algorithm to sample *a posteriori*. The latter approach is more robust and provides more knowledge of the *a posteriori* distribution than its optimization counterpart. We analyze our algorithm without assuming that the target distribution is strongly log-concave. Instead, we assume the weaker log Sobolev inequality, which allows for nonconvexity.

## 1 Introduction

Federated learning uses a set of techniques to efficiently distribute the training of a machine learning algorithm across several devices, who own the training data. In a federated learning environment, several considerations are considered, such as privacy or communication between the devices (Kairouz et al., 2021). In particular, the communication cost between the devices and a central server (Konečnỳ et al., 2016; McMahan et al., 2017) is one of the main bottlenecks in a federated learning environment. Indeed, classical training algorithms such as the Stochastic Gradient Descent (SGD) can formally be distributed between the server and the devices. But this requires communications between the devices and the server, raising communication concerns. Therefore, most federated learning algorithms rely on compressing the messages that are exchanged between the server and the devices. For instance, variants SGD involving compression (Karimireddy et al., 2020; Horváth et al., 2019b; Haddadpour et al., 2021; Das et al., 2020; Gorbunov et al., 2021) have successfully been applied in several engineering setups (Kairouz et al., 2021; Dimitriadis et al., 2022).

**Sampling in a federated learning environment.** Most of the literature on the training of federated learning learning algorithm focused on the optimization paradigm, i.e., the training algorithm is actually a communication efficient optimization algorithm to minimize the training loss $F$. But, optimization is not the only paradigm in the training of learning algorithms. The Bayesian approach relies on sampling from a distribution $\pi$ proportional to $\exp(-F)$ rather than minimizing $F$. In particular, the Bayesian approach provides a number of samples from $\exp(-F)$ rather than a single minimizer of $F$. The distribution $\pi$ is often the the posterior distribution of some Bayesian model. In this case, sampling from $\pi$ allows to perform some Bayesian computations such as computing confidence intervals.

In this paper, we take a Bayesian approach to federated learning, and similarly to Vono et al. (2022); Deng et al. (2021); El Mekkaoui et al. (2021), we focus on the communication complexity (we ignore privacy concerns). We introduce a communication efficient variant of the Langevin algorithm, a widely used algorithm to sample from a target distribution $\pi \propto \exp(-F)$ where $F$ is a smooth nonconvex function but $\pi$ satisfies the log-Sobolev inequality (Villani, 2009). We study the complexity of our communication efficient algorithm to sample from $\pi$ by exploiting the connections between optimization and sampling. More precisely, our approach has roots in the recent literature on viewing Langevin algorithm as an optimization algorithm over a space of probability distributions called the Wasserstein space (Wibisono, 2018; Durmus et al., 2019; Ambrosio et al., 2005).

## 1.1 Related Work

Most approaches to train federated learning algorithms rely on minimizing the training loss with a communication efficient variant of SGD. Several training algorithms have been proposed, but we mention Karimireddy et al. (2020); Horváth et al. (2019b); Haddadpour et al. (2021); Das et al. (2020); Gorbunov et al. (2021) because these papers contain the state of the art results in terms of minimization of the training loss. In this work we shall specifically use the MARINA gradient estimator introduced in Gorbunov et al. (2021) and inspired from Nguyen et al. (2017).

The literature on sampling with Langevin algorithm is also large. In recent years, the machine learning research community has specifically been interested in the complexity of Langevin algorithm (Dalalyan, 2017; Durmus & Moulines, 2017). In the case where $F$ is not convex, recent works include Vempala & Wibisono (2019); Wibisono (2019); Ma et al. (2019); Chewi et al. (2021) which use an isoperimetric-type inequality (Villani, 2009, Chapter 21) such as the log Sobolev inequality to prove convergence in Kullback-Leibler divergence. Other works on Langevin in the nonconvex case include Balasubramanian et al. (2022); Cheng et al. (2018); Majka et al. (2020); Mattingly et al. (2002).

The closest papers to our work are Vono et al. (2022); Deng et al. (2021). Like our work, these papers study the convergence of an efficient variant of Langevin algorithm for federated learning. The key difference between these papers and our work is that they assume $F$ strongly convex, whereas we only assume that the target distribution $\pi \propto \exp(-F)$ satisfies the log Sobolev inequality (LSI), which allows $F$ to be nonconvex. More precisely, the strong convexity of $F$ implies LSI (Villani, 2009, Chapter 21), but LSI allows $F$ to be nonconvex. We compare our complexity results to their in Table 1.

| Paper | Algorithm | Communication | Assumption |
|---|---|---|---|
| Vono et al. (2022) | QLSD$^{++}$ | **Reduced** | Strong convexity |
| Deng et al. (2021) | (FA-LD) | **Reduced** | Strong convexity |
| Wibisono (2019) | Langevin | Expensive | **log Sobolev inequality** |
| **This paper** | Langevin-Marina | **Reduced** | **log Sobolev inequality** |

Table 1: Langevin-Marina is the only algorithm in the list that works under LSI only, with a reduced communication complexity. The four algorithms in the list have the same complexity $\tilde{\mathcal{O}}\left(\frac{d}{\varepsilon^2}\right)$ to achieve $\varepsilon$ accuracy in 2-Wasserstein distance in dimension $d$.

Our contributions are summarized below.

### 1.2 Contributions

We consider the problem of sampling from $\pi \propto \exp(-F)$, where $\pi$ satisfies LSI, in a federated learning setup. We make the following contributions:

- We propose a communication efficient variant of Langevin algorithm, called Langevin-MARINA, that can be distributively implemented between the central server and the devices. Langevin-MARINA relies on compressed communications. More precisely, we borrow the MARINA estimator used in optimization to perform Sampling.

- We analyze the complexity of our sampling algorithm in terms of the Kullback-Leibler divergence, the Total Variation distance and the 2-Wasserstein distance. Our approach relies on viewing our sampling problem as an optimization problem over a space of probability measures, and allows $F$ to be nonconvex.

- Our sampling algorithm, Langevin-MARINA, is inspired from an optimization algorithm called MARINA (Gorbunov et al., 2021) for which we give a new convergence proof in Appendix B. This new proof draws connections between optimization (MARINA) and sampling (Langevin-MARINA).

### 1.3 Paper organization

In Section 2, we review some background material on sampling and optimization. Next, we introduce our federated learning setup in Section 3. In Section 4 we give our main algorithm, Langevin-MARINA, and our main complexity results. We conclude in Section 5. The proofs, including the new proofs of existing results for MARINA and the experiment results are deferred to the Appendix.

## 2 Preliminaries

### 2.1 Mathematical problem

Throughout this paper, we consider a nonconvex function $F : \mathbb{R}^d \to \mathbb{R}$ which is assumed to be $L$-smooth, i.e., differentiable with an $L$-Lipschitz continuous gradient: $\|\nabla F(x) - \nabla F(y)\| \leq L \|x - y\|$. As often in machine learning, $F$ can be seen as a training loss and takes the form of a finite sum

$$F = \sum_{i=1}^{n} F_i, \tag{1}$$

where each $F_i$ can be seen as the loss associated to the dataset stored in the device $i$. Assuming that $\int \exp(-F(x))dx \in (0, \infty)$, we denote by

$$\pi \propto \exp(-F), \tag{2}$$

the probability distribution whose density is proportional to $\exp(-F)$. We take a Bayesian approach: instead of minimizing $F$, our goal is to generate random samples from $\pi$, which can be seen as the posterior distribution of some Bayesian model.

### 2.2 Sampling and Optimal Transport

We denote by $\mathcal{P}_2(\mathbb{R}^d)$ the set of Borel measures $\sigma$ on $\mathbb{R}^d$ with finite second moment, that is $\int \|x\|^2 d\sigma(x) < +\infty$. For every $\sigma, \nu \in \mathcal{P}_2(\mathbb{R}^d)$, $\Gamma(\sigma, \nu)$ is the set of all the coupling measures between $\sigma$ and $\nu$ on $\mathbb{R}^d \times \mathbb{R}^d$, that is $\gamma \in \Gamma(\sigma, \nu)$ if and only if $\sigma(dx) = \gamma(dx, \mathbb{R}^d)$ and $\nu(dy) = \gamma(\mathbb{R}^d, dy)$. The Wasserstein distance between $\sigma$ and $\nu$ is defined by

$$W_2(\sigma, \nu) = \sqrt{\inf_{\gamma \in \Gamma(\sigma, \nu)} \int \|x - y\|^2 \gamma(dx, dy)}. \tag{3}$$

The Wasserstein distance $W_2(\cdot, \cdot)$ is a metric on $\mathcal{P}_2(\mathbb{R}^d)$ and the metric space $\left(\mathcal{P}_2(\mathbb{R}^d), W_2\right)$ is called the Wasserstein space, see Ambrosio et al. (2005).

The Kullback-Leibler (KL) divergence w.r.t. $\pi$ can be seen as the map from the Wasserstein space to $(0, \infty]$ defined for every $\sigma \in \mathcal{P}_2(\mathbb{R}^d)$ by

$$D_{\mathrm{KL}}(\sigma \mid \pi) := \begin{cases} \int \log(\frac{\sigma}{\pi})(x) d\sigma(x) & \text{if } \sigma \ll \pi \\ \infty & \text{else,} \end{cases} \tag{4}$$

where $\sigma \ll \pi$ means that $\sigma$ is absolutely continuous w.r.t. $\pi$ and $\frac{\sigma}{\pi}$ denotes the density of $\sigma$ w.r.t. $\pi$. The KL divergence is always nonnegative and it is equal to zero if and only if $\sigma = \pi$. Therefore, assuming that $\pi \in \mathcal{P}_2(\mathbb{R}^d)$, $\pi$ can be seen as the solution to the optimization problem

$$\min_{\sigma \in \mathcal{P}_2(\mathbb{R}^d)} D_{\mathrm{KL}}(\sigma \mid \pi). \tag{5}$$

Viewing $\pi$ as a minimizer of the KL divergence is the cornerstone of our approach. Indeed, we shall view the proposed algorithm as an optimization algorithm to solve Problem equation 5. In particular, following Wibisono (2018); Durmus et al. (2019), we shall view the Langevin algorithm as a first order algorithm over the Wasserstein space. In particular, the relative Fisher information, defined as

$$J_\pi(\sigma) := \begin{cases} \int \left\| \nabla \log(\frac{\sigma}{\pi})(x) \right\|^2 d\sigma(x) & \text{if } \sigma \ll \pi \\ \infty, & \text{else,} \end{cases} \tag{6}$$

will play the role of the squared norm of the gradient of the objective function. Indeed, by defining a differential structure over the Wasserstein space (Ambrosio et al., 2005), the relative Fisher information is the squared norm of the gradient of the KL divergence.

The analysis of nonconvex optimization algorithm minimizing $F$ often relies on a gradient domination condition relating the squared norm of the gradient to the function values, such as the following Lojasiewicz condition (Blanchet & Bolte, 2018; Karimi et al., 2016)

$$F(x) - \min F \leq \frac{1}{2\mu} \|\nabla F(x)\|^2. \tag{7}$$

The latter condition has a well-known analogue over the Wasserstein space for the KL divergence, called logarithmic Sobolev inequality.

**Assumption 2.1** (Logarithmic Sobolev Inequality (LSI)). The distribution $\pi$ satisfies the logarithmic Sobolev inequality with constant $\mu$: for all $\sigma \in \mathcal{P}_2(\mathbb{R}^d)$,

$$D_{\mathrm{KL}}(\sigma \mid \pi) \leq \frac{1}{2\mu} J_\pi(\sigma). \tag{8}$$

The log Sobolev inequality has been studied in the optimal transport community (Villani, 2009, Chapter 21) and holds for several target distributions $\pi$. First, if $F$ is $\mu$-strongly convex (we say that $\pi$ is strongly log-concave), then LSI holds with constant $\mu$. Besides, LSI is preserved under bounded perturbations and Lipschitz mappings (Vempala & Wibisono, 2019, Lemma 16, 19). Therefore, small perturbations of strongly-log concave distributions satisfy LSI. For instance, if we add to a Gaussian distribution another Gaussian distribution with a small weight, then the resulting mixture of Gaussian distributions is not log-concave but it satisfies LSI. One can also find compactly supported examples, see Wibisono (2019, Introduction).

LSI is a condition that has been used in the analysis of the Langevin algorithm

$$x_{k+1} = x_k - h\nabla F(x_k) + \sqrt{2h} Z_{k+1}, \tag{9}$$

where $h > 0$ is a step size and $(Z_k)$ a sequence of i.i.d standard Gaussian vectors over $\mathbb{R}^d$. Langevin algorithm can be seen as a gradient descent algorithm for $F$ to which a Gaussian noise is added at each step. For instance, Wibisono (2019) showed that the distributions $\rho_k$ of $x_k$ converges rapidly towards the target distribution $\pi$ in terms of the KL divergence under LSI. Other metrics have also been considered, such a the Total Variation distance

$$\|\sigma - \pi\|_{TV} = \sup_{A \in B(\mathbb{R}^d)} |\sigma(A) - \pi(A)|, \tag{10}$$

where $B(\mathbb{R}^d)$ denotes the Borel sigma field of $\mathbb{R}^d$.

---

**Algorithm 1** MARINA (Gorbunov et al., 2021)

---
1: **Input:** Starting point $x_0$, step-size $h$, number of iterations $K$
2: **for** $i = 1, 2, \cdots, n$ in parallel **do**
3:      Device $i$ computes MARINA estimator $g_0^i$
4:      Device $i$ uploads $g_0^i$ to the central server
5: **end for**
6: Server aggregates $g_0 = \frac{1}{n} \sum_{i=1}^n g_0^i$
7: **for** $k = 0, 1, 2, \cdots, K - 1$ **do**
8:      Server broadcasts $g_k$ to all devices $i$
9:      **for** $i = 1, 2, \cdots, n$ in parallel **do**
10:          Device $i$ performs $x_{k+1} = x_k - hg_k$
11:          Device $i$ computes MARINA estiamtor $g_{k+1}^i$
12:          Device $i$ uploads $g_{k+1}^i$ to the central server
13:      **end for**
14:      Server aggregates $g_{k+1} = \frac{1}{n} \sum_{i=1}^n g_{k+1}^i$
15: **end for**
16: **Return:** $\hat{x}^K$ chosen uniformly at random from $(x_k)_{k=0}^K$ or last point $x_K$

---

## 3 Federated learning

### 3.1 Example of a federated learning algorithm: MARINA

We now describe our federated learning setup. A central server and a number of devices are required to run a training algorithm in a distributed manner. Each device $i \in \{1, \ldots, n\}$ owns a dataset. A meta federated learning algorithm is as follows: at each iteration, each device performs some local computation (for instance, a gradient computation) using its own dataset, then each device sends the result of that computation to the central server. The central server aggregates the results and sends that aggregation back to the devices. Algorithm 1 provides an example of a federated learning algorithm. This algorithm, called MARINA, was introduced and analyzed in Gorbunov et al. (2021) under the assumption that the sequence $(g_k)$ is a MARINA estimator of the gradient as defined below.

**Definition 3.1.** Given a sequence produced by an algorithm $(x_k)_k$, a MARINA estimator of the gradient is a random sequence $(g_k)_k$ satisfying $\mathrm{E}[g_k] = \mathrm{E}[\nabla F(x_k)]$ and

$$\mathbf{G}_{k+1} \le (1-p)\mathbf{G}_k + (1-p)L^2\alpha\mathrm{E}\left[\|x_{k+1} - x_k\|^2\right] + \theta, \tag{11}$$

where $\mathbf{G}_k = \mathrm{E}\left[\|g_k - \nabla F(x_k)\|^2\right]$ and $0 < p \le 1$, $\alpha, \theta \ge 0$.

MARINA iterations can be rewritten as

$$x_{k+1} = x_k - hg_k. \tag{12}$$

Since $\mathrm{E}[g_k] = \mathrm{E}[\nabla F(x_k)]$, MARINA can be seen as a SGD distributed between the central server and the devices, to minimize the nonconvex function $F$.

Recall that in federated learning, the main bottleneck is the communication cost between server and devices. The key aspect of MARINA is that it achieves the state of the art in terms of communication complexity among nonconvex optimization algorithms. The reason is the following: there are practical examples of MARINA estimators $(g_k)$ that rely on a compression step before the communication and are therefore cheap to communicate[1].

---

[1]Note that we did not specify how $g_k^i$ is computed in Algorithm 1. For the moment we only require $g_k$ to be a MARINA estimator

### 3.2 Compression operator

MARINA estimators are typically cheaper to communicate than full gradient because they involve a compression step, making the communication light. A compression operator is a random map from $\mathbb{R}^d$ to $\mathbb{R}^d$ with the following properties.

**Definition 3.2** (Compression). A stochastic mapping $\mathcal{Q} : \mathbb{R}^d \to \mathbb{R}^d$ is a compression operator if there exists $\omega > 0$ such that for any $x \in \mathbb{R}^d$,

$$\mathrm{E}\left[\mathcal{Q}(x)\right] = x, \quad \mathrm{E}\left[\|\mathcal{Q}(x) - x\|^2\right] \leq \omega\|x\|^2. \tag{13}$$

The first equation states that $\mathcal{Q}$ is unbiased and the second equation states that the variance of the compression operator has a quadratic growth. There are lots of compression operators satisfying equation 13, explicit examples based on quantization and/or sparsification are given in Alistarh et al. (2017); Horváth et al. (2019a). In general, compressed quantities are cheap to communicate.

We now fix a compression operator and give examples of gradient estimators involving compression, which are provably MARINA estimators.

### 3.3 MARINA gradient estimators

In every examples presented below, $g_k := \frac{1}{n}\sum_{i=1}^n g_k^i$, where $g_k^i$ is meant to approximate $\nabla F_i(x_k)$, i.e., the gradient at the device $i$, at step $k$. We consider any random sequence $(x_k)$ generated by a training algorithm.

#### 3.3.1 Vanilla MARINA gradient estimator

The vanilla MARINA gradient estimator is defined as the average $g_k := \frac{1}{n}\sum_{i=1}^n g_k^i$, where for every $i \in \{1, \ldots, n\}$, $g_0^i = \nabla F_i(x_0)$ and

$$g_{k+1}^i := \begin{cases} \nabla F_i(x_{k+1}) & \text{prob. } p > 0 \\ g_k + \mathcal{Q}\left(\nabla F_i(x_{k+1}) - \nabla F_i(x_k)\right) & \text{prob. } 1 - p \end{cases}. \tag{14}$$

Node $i$ randomly (and independently of $\mathcal{Q}, g_k, x_{k+1}$) computes and uploads to the server the gradient of $F_i$ at point $x_{k+1}$ with probability $p$. Else, node $i$ computes and uploads the compressed difference of the gradients of $F_i$ between points $x_{k+1}$ and $x_k$: $\mathcal{Q}\left(\nabla F_i(x_{k+1}) - \nabla F_i(x_k)\right)$. Note that the server already knows $g_k$ from the previous iteration, so no need to send $g_k$ to the server. In average, device $i$ computes a full gradient every $1/p$ steps ($p$ is very small, see Remark 3.6 below).

**Proposition 3.3.** *Assume that $F_i$ is $L_i$-smooth for all $i \in \{1, \ldots, n\}$, i.e.,*

$$\|\nabla F_i(x) - \nabla F_i(y)\| \leq L_i\|x - y\|, \quad \forall x, y \in \mathbb{R}^d. \tag{15}$$

*Then, $g_k = \frac{1}{n}\sum_{i=1}^n g_k^i$, where $g_k^i$ is defined by equation 14 is a MARINA estimator in the sense of Definition 3.1, with $\alpha = \frac{\omega \sum_{i=1}^n L_i^2}{n^2 L^2}, \theta = 0$.*

This proposition means that the MARINA estimator reduces the variance induced by the compression operator. Note that, if $F_i$ is $L_i$-smooth, then $L \leq \frac{1}{n}\sum_{i=1}^n L_i$.

#### 3.3.2 Finite sum case

Consider the finite sum case where each $F_i$ is a sum over the data points stored in the device $i$: $F_i = \sum_{j=1}^N F_{ij}$, where $F_{ij}$ is the loss associated to the data point $j$ in the device $i$. The finite sum MARINA gradient estimator is analogue to the vanilla estimator, but with subsampling. It is defined as the average $g_k := \frac{1}{n}\sum_{i=1}^n g_k^i$, where for every $i \in \{1, \ldots, n\}$, $g_0^i = \nabla F_i(x_0)$ and

$$g_{k+1}^i := \begin{cases} \nabla F_i(x_{k+1}) & \text{prob. } p > 0 \\ g_k + \mathcal{Q}\left(\frac{1}{b'}\sum_{j \in I'_{i,k}}\left(\nabla F_{ij}(x_{k+1}) - \nabla F_{ij}(x_k)\right)\right) & \text{prob. } 1 - p \end{cases}, \tag{16}$$

where $b'$ is the minibatch size and $I'_{i,k}$ is the set of the indices in the minibatch, $\left|I'_{i,k}\right| = b'$. Here, $(I'_{i,k})_{i,k}$ are i.i.d random sets consisting in $b'$ i.i.d samples from the uniform distribution over $\{1, \ldots, N\}$.

**Proposition 3.4.** *Assume that $F_{ij}$ is $L_{ij}$-smooth for all $i \in \{1, \ldots, n\}$ and all $j \in \{1, \ldots, N\}$.*

*Then, $g_k = \frac{1}{n} \sum_{i=1}^{n} g_k^i$, where $g_k^i$ is defined by equation 16 is a* MARINA *estimator in the sense of Definition 3.1, with $\alpha = \frac{\omega \sum_{i=1}^{n} L_i^2 + (1+\omega) \frac{\sum_{i=1}^{n} \mathcal{L}_i^2}{b'}}{n^2 L^2}$, where $\mathcal{L}_i = \max_{j \in [N]} L_{ij}$ and $\theta = 0$.*

This proposition means that the MARINA estimator reduces the variance induced by the compression operator and the variance induced by the subsampling.

### 3.3.3 Online case

Consider the online case where each $F_i$ is an expectation over the randomness of a stream of data arriving online: $F_i(x) = \mathbb{E}_{\xi_i \sim \mathcal{D}_i} [F_{\xi_i}(x)]$, where $F_{\xi_i}$ is the loss associated to a data point $\xi_i$ in device $i$. The online MARINA gradient estimator is analogue to the finite sum estimator, but with subsampling from $\mathcal{D}_i$ instead of the uniform over $\{1, \ldots, N\}$. Moreover, the full gradient $\nabla F_i$ is never computed (because it is intractable in online learning). The online MARINA gradient estimator is defined as the average $g_k := \frac{1}{n} \sum_{i=1}^{n} g_k^i$, where for every $i \in \{1, \ldots, n\}$, $g_0^i = \frac{1}{b} \sum_{\xi \in I_{i,0}} \nabla F_\xi(x_0)$ and

$$g_{k+1}^i := \begin{cases} \frac{1}{b} \sum_{\xi \in I_{i,k}} \nabla F_\xi(x_{k+1}) & \text{prob. } p > 0 \\ g_k + \mathcal{Q}\left(\frac{1}{b'} \sum_{\xi \in I'_{i,k}} (\nabla F_\xi(x_{k+1}) - \nabla F_\xi(x_k))\right) & \text{prob. } 1 - p \end{cases}, \tag{17}$$

where $I'_{i,k}$ and $I_{i,k}$ are the set of the indices in a minibatch, $\left|I'_{i,k}\right| = b'$ and $|I_{i,k}| = b$. Here, $(I_{i,k})_{i,k}$ (resp. $(I'_{i,k})_{i,k}$) is an i.i.d random set consisting in $b$ (resp. $b'$) i.i.d samples from $\mathcal{D}_i$. Besides, in the online case only, we make a bounded gradient assumption.

**Proposition 3.5.** *Assume that for all $i \in \{1, \ldots, n\}$ there exists $\sigma_i \geq 0$ such that for all $x \in \mathbb{R}^d$,*

$$\mathbb{E}_{\xi_i \sim \mathcal{D}_i} [\nabla F_{\xi_i}(x)] = \nabla F_i(x),$$
$$\mathbb{E}_{\xi_i \sim \mathcal{D}_i} \left[\|\nabla F_{\xi_i}(x) - \nabla F_i(x)\|^2\right] \leq \sigma_i^2.$$

*Moreover, assume that $F_{\xi_i}$ is $L_{\xi_i}$-smooth $\mathcal{D}_i$ almost surely. Then, $g_k = \frac{1}{n} \sum_{i=1}^{n} g_k^i$, where $g_k^i$ is defined by equation 17 is a* MARINA *estimator in the sense of Definition 3.1, with $\alpha = \frac{\omega \sum_{i=1}^{n} L_i^2 + (1+\omega) \frac{\sum_{i=1}^{n} \mathcal{L}_i^2}{b'}}{n^2 L^2}$, where $\mathcal{L}_i$ is the $L^\infty$ norm of $L_{\xi_i}$ (where $\xi_i \sim \mathcal{D}_i$) and $\theta = \frac{p \sum_{i=1}^{n} \sigma_i^2}{n^2 b}$.*

Note that in the online case $\theta \neq 0$ in general, because the online MARINA estimator does not reduce to zero the variance induced by the subsampling. That is because the loss $F_i$ is an expectation (and not a finite sum *a priori*), for which variance reduction to zero is impossible in general. However, $\theta$ can be made small by taking a large minibatch size $b$ or a small probability $p$. Moreover, in the online case, $\mathbf{G}_0 \neq 0$ unlike in the two other cases.

*Remark* 3.6. Typically, $p$ is very small, for instance, Gorbunov et al. (2021) chooses $p = \zeta_\mathcal{Q}/d$ in equation 14, $p = \min\{\zeta_\mathcal{Q}/d, b'/(N+b')\}$ in equation 16 and $p = \min\{\zeta_\mathcal{Q}/d, b'/(b+b')\}$ in equation 17, where $\zeta_\mathcal{Q} = \sup_{x \in \mathbb{R}^d} \mathbb{E}[\|\mathcal{Q}(x)\|_0]$ and $\|y\|_0$ is the number of non-zero components of $y \in \mathbb{R}^d$. Therefore, with high probability $1 - p$, the compressed difference $\mathcal{Q}(\nabla F_i(x_{k+1}) - \nabla F_i(x_k))$ is sent to the server. Sending compressed quantities has a low communication cost (Alistarh et al., 2017; Horváth et al., 2019a).

*Remark* 3.7. Propositions 3.3, 3.4 and 3.5 can be found in Gorbunov et al. (2021, Equation 21, 33, 46) in the case where $(x_k)$ is a stochastic gradient descent algorithm (i.e., $x_{k+1} = x_k - hg^k$). In the general case where $(x_k)$ is produced by any algorithm, we found out that the proofs of these Propositions are the same, but we reproduce these proofs in the Appendix for the sake of completeness.

## 4  Langevin-MARINA

In this section, we give our main algorithm, Langevin-MARINA, and study its convergence for sampling from $\pi \propto \exp(-F)$. We prove convergence bounds in KL divergence, 2-Wasserstein distance and Total Variation distance under LSI.

### 4.1  Proposed algorithm

Our main algorithm, Langevin-MARINA can be seen as a Langevin variant of MARINA which adds a Gaussian noise at each step of MARINA (Algorithm 1). Our motivation is to obtain a Langevin algorithm whose communication complexity is similar to that of MARINA. Alternatively, one can see Langevin-MARINA as a MARINA variant of Langevin algorithm which uses a MARINA estimator of the gradient $g_k = \frac{1}{n} \sum_{i=1}^{n} g_k^i$ instead of the full gradient $\nabla F(x_k)$ in the Langevin algorithm:

$$x_{k+1} = x_k - hg_k + \sqrt{2h}Z_{k+1}. \tag{18}$$

Langevin-MARINA is presented in Algorithm 2.

---

**Algorithm 2** Langevin-MARINA (proposed algorithm)

---

1:  **Input:** Starting point $x_0 \sim \rho_0$, step-size $h$, number of iterations $K$
2:  **for** $i = 1, 2, \cdots, n$ in parallel **do**
3:      Device $i$ computes MARINA estimator $g_0^i$
4:      Device $i$ uploads $g_0^i$ to the central server
5:  **end for**
6:  Server aggregates $g_0 = \frac{1}{n} \sum_{i=1}^{n} g_0^i$
7:  **for** $k = 0, 1, 2, \cdots, K-1$ **do**
8:      Server broadcasts $g_k$ to all devices $i$
9:      Server draws a Gaussian vector $Z_{k+1} \sim \mathcal{N}(0, I_d)$
10:      Server performs $x_{k+1} = x_k - hg_k + \sqrt{2h}Z_{k+1}$
11:      Server broadcasts $x_{k+1}$ to all devices $i$
12:      **for** $i = 1, 2, \cdots, n$ in parallel **do**
13:          Device $i$ computes MARINA estimator $g_{k+1}^i$
14:          Device $i$ uploads $g_{k+1}^i$ to the central server
15:      **end for**
16:      Server aggregates $g_{k+1} = \frac{1}{n} \sum_{i=1}^{n} g_{k+1}^i$
17:  **end for**
18:  **Return:** $x_K$

---

### 4.2  Communication complexity of Langevin-MARINA

Compared to MARINA where $n$ equivalent stochastic gradient descent steps are performed by the devices, here the Langevin step is performed only once, by the server. This allows for computation savings ($n$ times less computations) at the cost of more communication: the iterates $x_k$ need to be broadcast by the server to the devices. Therefore, the communication complexity of the sampling algorithm Langevin-MARINA is higher than the communication complexity of the optimization algorithm MARINA.

However, compared to the concurrent sampling algorithm of Vono et al. (2022), the communication complexity per iteration of Langevin-MARINA is equivalent. Comparing the communication complexity of Langevin-MARINA to that of FA-LD (Deng et al., 2021) is more difficult because FA-LD makes communication savings by performing communication rounds only after a number $T$ of local updates, instead of using a compression operator after each local update like Langevin-MARINA.

### 4.3  Convergence theory for Langevin-MARINA

We first prove the convergence of Langevin-MARINA in KL-divergence.

**Theorem 4.1.** *Assume that LSI (Assumption 2.1) holds and that $g_k = \frac{1}{n}\sum_{i=1}^{n} g_k^i$ is a MARINA estimator in the sense of Definition 3.1. If*

$$0 < h \le \min\left\{\frac{1}{14L}\sqrt{\frac{p}{1+\alpha}}, \frac{p}{6\mu}\right\}, \tag{19}$$

*then*

$$D_{\text{KL}}\left(\rho_K \mid \pi\right) \le e^{-\mu Kh}\Psi_3 + \frac{1 - e^{-K\mu h}}{\mu}\tau, \tag{20}$$

*where* $\Psi_3 = D_{\text{KL}}\left(\rho_0 \mid \pi\right) + \frac{1-e^{-\mu h}}{\mu}C\boldsymbol{G}_0, \tau = \left(2L^2 + C(1-p)L^2\alpha\right)\left(8Lh^2d + 4dh\right) + C\theta, C = \frac{8L^2h^2\beta + 2\beta}{1-(1-p)(4L^2h^2\alpha+1)\beta}, \beta = e^{\mu h}$ *and $\rho_k$ is the distribution of $x_k$ for every $k$.*

*In particular, if $\theta = 0$, set $h = \mathcal{O}\left(\frac{\mu p\varepsilon}{L^2(1+\alpha)d}\right)$, $K = \Omega\left(\frac{L^2(1+\alpha)d}{\mu^2 p\varepsilon}\log\left(\frac{\Psi_3}{\varepsilon}\right)\right)$ then $D_{\text{KL}}\left(\rho_K \mid \pi\right) \le \varepsilon$. If $\theta \ne 0$, there will always be an extra residual term $\frac{1-e^{-K\mu h}}{\mu}C\theta$ in the right hand side of equation 20 which cannot be diminished to 0 by setting $K$ and $h$. However in the online case (Section 3.3.3), $\theta = \frac{p\sum_{i=1}^{n}\sigma_i^2}{n^2 b}$, we can set $b = \Omega\left(\frac{\sum_{i=1}^{n}\sigma_i^2}{\mu n^2\varepsilon}\right)$ to make the residual term small $\frac{1-e^{-K\mu h}}{\mu}C\theta = \mathcal{O}(\varepsilon)$.*

From this theorem, we can obtain complexity results in Total Variation distance and 2-Wasserstein distance. Indeed, Pinsker inequality states that

$$\|\sigma - \pi\|_{TV} \le \sqrt{\frac{1}{2}D_{\text{KL}}\left(\sigma \mid \pi\right)}, \forall \sigma, \nu \in \mathcal{P}_2(\mathbb{R}^d), \tag{21}$$

and LSI implies Talagrand's $T_2$ inequality

$$W_2^2(\sigma, \pi) \le \frac{2}{\mu}D_{\text{KL}}\left(\sigma \mid \pi\right), \forall \sigma \in \mathcal{P}_2(\mathbb{R}^d), \tag{22}$$

see Villani (2009, Chapter 21).

**Corollary 4.2.** *Let assumptions and parameters be as in Theorem 4.1. Then,*

$$\|\rho_K - \pi\|_{TV}^2 \le \frac{1}{2}\left(e^{-\mu Kh}\Psi_3 + \frac{1 - e^{-K\mu h}}{\mu}\tau\right), \tag{23}$$

*and*

$$W_2^2(\rho_K, \pi) \le \frac{2}{\mu}\left(e^{-\mu Kh}\Psi_3 + \frac{1 - e^{-K\mu h}}{\mu}\tau\right). \tag{24}$$

In particular, if $\theta = 0$, set $h = \mathcal{O}\left(\frac{\mu p\varepsilon^2}{L^2(1+\alpha)d}\right)$, $K = \Omega\left(\frac{L^2(1+\alpha)d}{\mu^2 p\varepsilon^2}\log\left(\frac{\Psi_3}{\varepsilon^2}\right)\right)$ then $\|\rho_K - \pi\|_{TV} \le \varepsilon$. If $\theta \ne 0$, there will always be an extra residual term $\frac{1-e^{-K\mu h}}{\mu}C\theta$ in the right hand side of equation 23 which cannot be diminished to 0 by setting $K$ and $h$. However in the online case (Section 3.3.3), $\theta = \frac{p\sum_{i=1}^{n}\sigma_i^2}{n^2 b}$, we can set $b = \Omega\left(\frac{\sum_{i=1}^{n}\sigma_i^2}{\mu n^2\varepsilon^2}\right)$ to make the residual term small $\frac{1-e^{-K\mu h}}{\mu}C\theta = \mathcal{O}(\varepsilon^2)$.

Finally, if $\theta = 0$, set $h = \mathcal{O}\left(\frac{\mu^2 p\varepsilon^2}{L^2(1+\alpha)d}\right)$, $K = \Omega\left(\frac{L^2(1+\alpha)d}{\mu^3 p\varepsilon^2}\log\left(\frac{\Psi_3}{\mu\varepsilon^2}\right)\right)$ then $W_2(\rho_K, \pi) \le \varepsilon$. If $\theta \ne 0$, there will always be an extra residual term $\frac{1-e^{-K\mu h}}{\mu^2}C\theta$ in the right hand side of equation 24 which cannot be diminished to 0 by setting $K$ and $h$. However in the online case, $\theta = \frac{p\sum_{i=1}^{n}\sigma_i^2}{n^2 b}$, we can set $b = \Omega\left(\frac{\sum_{i=1}^{n}\sigma_i^2}{\mu^2 n^2\varepsilon^2}\right)$ to make the residual term small $\frac{1-e^{-K\mu h}}{\mu^2}C\theta = \mathcal{O}(\varepsilon^2)$.

We can compare our results to Vono et al. (2022); Deng et al. (2021) which also obtain bounds in 2-Wasserstein distance under the stronger assumption that $F$ is $\mu$-strongly convex, see Table 1. In Vono et al.

(2022), they need $h = \mathcal{O}\left(\frac{\varepsilon^2}{dl}\right)$ and $K = \Omega\left(\frac{dl}{\varepsilon^2}\log\left(\frac{W_2^2(\rho_0,\pi)}{\varepsilon^2}\right)\right)$, where they compute the full gradient every $l$ step (we do not include the dependence in $\mu, L$ in their result for the sake of simplicity). One can think of $l$ as $1/p$. In Deng et al. (2021), they require $h = \mathcal{O}\left(\frac{\mu^2\varepsilon^2}{dL^2(T^2\mu+L)}\right)$ and $K = \Omega\left(\frac{dL^2(T^2\mu+L)}{\mu^3\varepsilon^2}\log\left(\frac{d}{\varepsilon^2}\right)\right)$ to get $W_2(\rho_K,\pi)\le\varepsilon$, where $T$ denotes the number of local updates.

*Remark* 4.3. Here, we provide insights into determining the optimal choice for the parameter $p$ and subsequently evaluate the associated communication cost. For the compression operator, we choose the $\text{Top}_k$ compression operator from Stich et al. (2018), for this compression operator, we have $\omega = 1 - s$, where $s := \frac{k}{d} \in [0,1]$. We only consider the vanilla MARINA gradient estimator case (thus $\theta = 0$), the other case can be considered similarly. Remember the formula for $\tau, C$ in Theorem 4.1 and $\alpha$ in Proposition 3.3, we can estimate that $\alpha = \frac{1-s}{n_l}$, where $n_l := \frac{n^2 L^2}{\sum_{i=1}^n L_i^2}$, $\tau \approx L^2(1 + \frac{(1-p)(1-s)}{pn_l})(8Lh^2d + 4dh)$, so to have $\frac{1-e^{K\mu h}}{\mu}\tau \le \epsilon^2$, we need $h = \tilde{\mathcal{O}}(\frac{\epsilon^2/d}{1+\frac{(1-p)(1-s)}{n_l p}})$, where we omit the dependence on $\mu, L$; to have $\frac{2}{\mu}e^{-\mu K h}\Psi_3 < \epsilon^2$, we need $K = \tilde{\Omega}(\frac{d(1+\frac{(1-p)(1-s)}{n_l p})}{\epsilon^2})$, where we omit the log term and dependence on $\mu, L$. With $K$, the communication complexity is propotional to $\frac{d(1+\frac{(1-p)(1-s)}{n_l p})}{\epsilon^2}(p + (1-p)s)$, and from it we can derive the optimal $p$ equals $p_k := \sqrt{\frac{s}{n_l+s}}$ and thus the optimal communication cost is of order $\tilde{\Omega}(\frac{d(1+\frac{(1-p_k)(1-s)}{n_l p_k})}{\epsilon^2}(p_k + (1-p_k)s))$ for $\text{Top}_k$ compression operator. It is hard to derive the optimal $k$ from this complicated formula, however once we have an estimate of $n_l$, we can approximate the optimal $k$ numerically by optimization algorithm. Since $s := \frac{k}{d} \in [0,1]$ and $n_l \approx n$ ( if the variance of $\{L_i\}_{i=1}^n$ is not big); so we have $p_k \approx \sqrt{\frac{k}{nd}}$, and so the communication cost is $\tilde{\Omega}(\frac{(k+\sqrt{\frac{dk}{n}})(1+\sqrt{\frac{d}{nk}})}{\epsilon^2})$ (so the best $k = 1$ and the corresponding $p \approx \sqrt{\frac{1}{nd}}$). For the algorithm in Vono et al. (2022), since the algorithm use compression operator in each step, so the communication complexity is of order $\tilde{\Omega}(\frac{dl}{\epsilon^2}\frac{k}{d}) = \tilde{\Omega}(\frac{kl}{\epsilon^2})$ (so the best $k = 1$) and similarly the algorithm in Deng et al. (2021) has communication complexity $\tilde{\Omega}(\frac{dL^2(T\mu+\frac{L}{T})}{\mu^3\epsilon^2})$ (so the best $T = \sqrt{\frac{L}{\mu}}$), since it only communicate with the central server without compression after every $T$ local updates.

Regarding the original MARINA algorithm for optimization, our proofs for Langevin-MARINA employ a dynamical point of view: we consider a path $\{x_t\}$ that goes from $x_k$ to $x_{k+1}$ and we bound the derivative of $F(x_t)$ along this path, this challenge is addressed in a novel manner by combining Lemma C.2 in the appendix and the properties of MARINA estimator, later we integrate along this path to bound the objective gap. Compared with existing federated sampling approaches, Langevin-MARINA does not rely on the strong convexity of $F$, and because of this lack of convexity we cannot hope to bound the distance between the iterate and the solution in Wasserstein distance directly. Therefore, we make a detour to first bound the KL divergence, which plays the role of an objective gap.

## 5 Conclusion

We introduced a communication efficient variant of Langevin algorithm for federated learning called Langevin-MARINA. We studied the complexity of this algorithm in terms of KL divergence, Total Variation distance and 2-Wasserstein distance. Unlike existing works on sampling for federated learning, we only require the target distribution to satisfy LSI, which allows the target distribution to not be log-concave.

Langevin-MARINA is inspired by a optimization algorithm for federated learning called MARINA. MARINA achieves the state of the art in communication complexity among nonconvex optimization algorithms. However, Langevin-MARINA requires more communication rounds than MARINA. The fundamental reason for this is that Langevin-MARINA needs to communicate the Gaussian noise. To solve this issue, one approach is to design a Langevin algorithm allowing for the compression of the Gaussian noise. Another approach is to put the same random seed in each device in order to ensure that they all generate the same $Z_{k+1}$ at step $k$. This approach does not require the communication of the Gaussian noise.

Finally, simple experiments have shown similar performance for Langevin-Marina and QLSD$^{++}$ Vono et al. (2022), although the convergence of QLSD$^{++}$ under LSI has not been established. Therefore, we wonder if a convergence theory of QLSD$^{++}$ under LSI would be possible. We leave this question for future work.

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

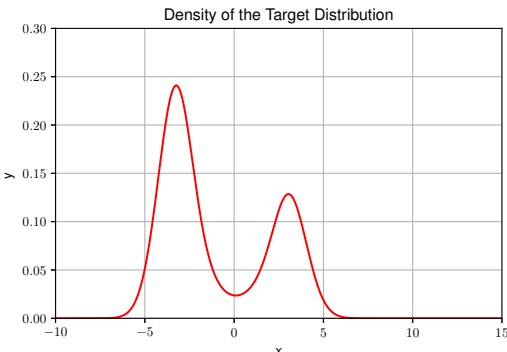

Figure 1: Density of $\pi$.

# A  Experiments

We now experiment our algorithm and compare it to QLSD$^{++}$ (Vono et al., 2022). Since this paper is theoretical, we chose a simple one dimensional example, in order to be able to visualize the performance of the algorithms on histograms.

The target distribution is $\pi \propto \exp(-F)$, where

$$F(x) = \begin{cases} \frac{(x+\Pi)^2}{2} + \frac{x}{10} & x \leq -\Pi \\ \cos(x) + 1 + \frac{x}{10} & -\Pi \leq x \leq \Pi \\ \frac{(x-\Pi)^2}{2} + \frac{x}{10} & x \geq \Pi \end{cases}, \tag{25}$$

and where $\Pi$ is the area of the disc with unit radius. The density of the target distribution is plotted in Figure 1. We can observe that $F$ is $L$-smooth but not convex. Moreover, we observe that $\pi$ is a bounded perturbation of a strongly log concave distribution. Therefore, $\pi$ satisfies LSI (with an unknown constant).

**Experimental setup.**  We use $n = 5$ clients, and client $i$ has $N(i)$ subfunctions $F_{ij}$. The number $N(i)$ is randomly chosen between 10 and 20. The subfunctions $F_{ij}$ are written $F_{ij}(x) = F(x) + \xi_{ij} \times \sin(x)$, where $\xi_{ij}$ is a noise factor such that $F_i(x) := \frac{1}{N(i)} \sum_{j=1}^{N(i)} F_{ij}(x)$. We compare Langevin-Marina to QLSD$^{++}$ (Vono et al., 2022). In Langevin-Marina we set $p = 0.001$, i.e., a full gradient is computing every 1000 iterations in expectation. In QLSD$^{++}$ (Vono et al., 2022) we set $\alpha = 0.2$, the initial memory term $\eta(i) = 0, i \in [5]$ and $l = 1000$, i.e., a full gradient is computing every 1000 iterations. We use the compression operator from Vono et al. (2022) with quantization level $s = 2^8$. For the ease of comparison, we also compress the $p$-branch in Langevin-Marina, so the communication complexity of Langevin-Marina and QLSD$^{++}$ are the same. To compute the stochastic gradient we subsample one index from $\{1, \ldots, N(i)\}$ in each iteration. We run 400000 steps of both algorithms and collect all the points generated by each algorithm to plot their histogram. The results are listed in Figure 2.

On Figure 2, the performance of the algorithms can be seen visually if the histogram (in blue) fits the curve (in red). Langevin-Marina and QLSD$^{++}$ have similar performances in this one-dimensional experiment, although the convergence of QLSD$^{++}$ under LSI has not been established, unlike Langevin-Marina. Therefore, we wonder if a convergence theory of QLSD$^{++}$ under LSI would be possible.

# B  New proofs of existing optimization results for MARINA (Algorithm 1)

The goal of this section is to we show that the approach that we used to study Langevin-MARINA can be used to study MARINA. **This section is, with the proof section Appendix C.6, the ONLY section of the paper that deals with the problem of minimizing $F$ rather than sampling $\exp(-F)$.**

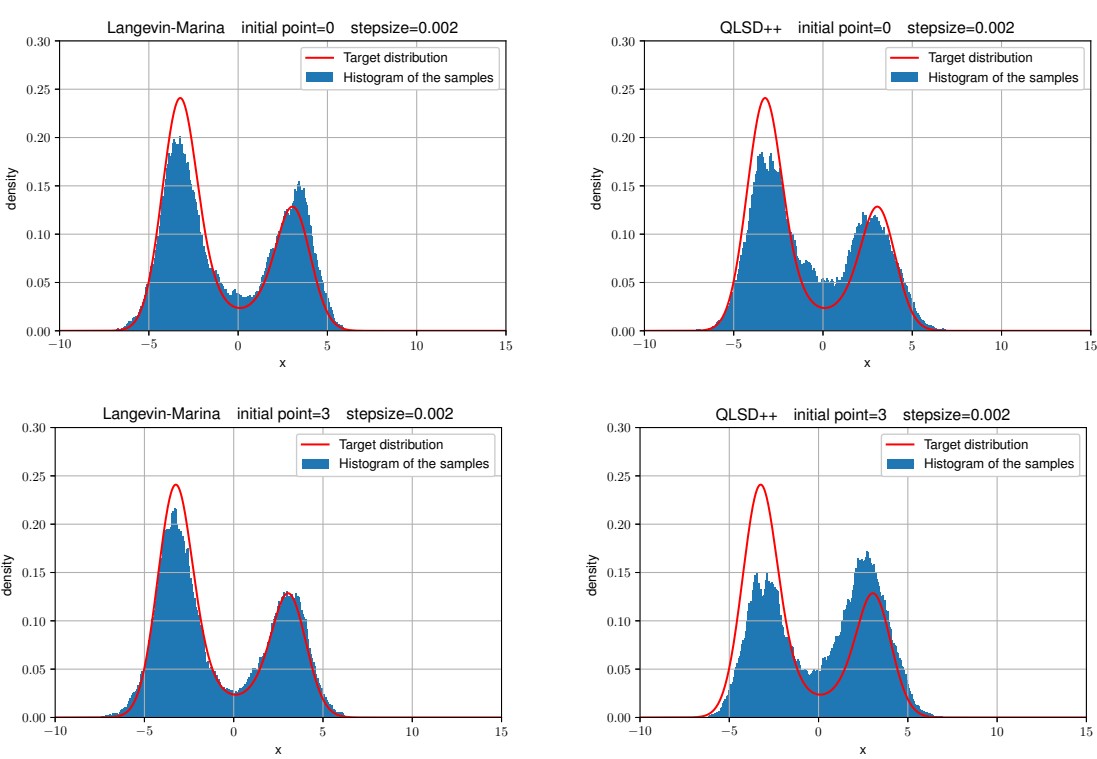

Figure 2: Langevin-Marina vs QLSD$^{++}$ with the same step-size $\gamma = 0.002$, but different initial point $x_0 = 0$ and $x_0 = 3$.

In particular, we provide a new analysis of MARINA (Algorithm 1) establishing the convergence of the whole continuous trajectories $(x_t)_{t\geq 0}$ generated by the algorithm, unlike Gorbunov et al. (2021) which focuses on the discrete iterates $(x_k)_{k=0}^K$. Compared to Gorbunov et al. (2021), we obtain the same convergence rate, but with a different method.

The approach we take to study MARINA is similar to the approach we took to study Langevin-MARINA: we establish the convergence of the whole continuous trajectories. The key difference is that MARINA is an optmization algorithm in the Euclidean space $\mathbb{R}^d$, so the underlying space in the convergence proofs of MARINA is $\mathbb{R}^d$. On the contrary, we viewed Langevin-MARINA as an optimization algorithm in the Wasserstein space, so the underlying space in the convergence proofs of Langevin-MARINA is the Wasserstein space.

In this section, we focus on the minimization, by MARINA (Algorithm 1), of the empirical risk:

$$\min_{x\in\mathbb{R}^d} F(x) = \sum_{i=1}^n F_i(x). \tag{26}$$

One can see MARINA as a variant of the gradient descent algorithm which uses a MARINA estimator of the gradient $g_k = \frac{1}{n}\sum_{i=1}^n g_k^i$ instead of the full gradient $\nabla F(x_k)$ in the gradient descent algorithm:

$$x_{k+1} = x_k - hg_k. \tag{27}$$

MARINA is presented in Algorithm 1.

We now provide the convergence results of MARINA. The proofs are provided later in the Appendix.

**Theorem B.1.** *Let $(g_k)_{k=0}^{K-1}$ be MARINA estimators. If the step-size satisfies*

$$0 < h \leq \frac{1}{10L}\sqrt{\frac{p}{1+\alpha}}, \tag{28}$$

*then*

$$\frac{1}{Kh}\int_0^{Kh} \mathrm{E}\left[\|\nabla F(x_t)\|^2\right] dt \leq \frac{2\left(\Psi_1 - F(x^*)\right)}{Kh} + 2C\theta, \tag{29}$$

*where $\Psi_1 = F(x_0) + hC\,\boldsymbol{G}_0, C = \frac{8L^2h^2+2}{1-(1-p)(4L^2h^2\alpha+1)}, x_t := x_{\lfloor\frac{t}{h}\rfloor h} + (t - \lfloor\frac{t}{h}\rfloor h)x_{\lceil\frac{t}{h}\rceil h}$.*

Let $h = \frac{1}{10L}\sqrt{\frac{p}{1+\alpha}}$ (hence $C = \mathcal{O}\left(\frac{1}{p}\right)$) and $\hat{x}_t := x_T$, where $T$ is a uniform random variable over $[0, Kh]$ independent of $(x_t)$. Then, $\mathrm{E}\left[\|\nabla F(\hat{x}_t)\|^2\right] = \frac{1}{Kh}\int_0^{Kh}\mathrm{E}\left[\|\nabla F(x_t)\|^2\right]dt$. To achieve $\mathrm{E}\left[\|\nabla F(\hat{x}_t)\|^2\right] \leq \varepsilon^2$ when $\theta = 0$ (for instance for the MARINA estimators equation 14 and equation 16), $\Omega\left(\frac{(\Psi_1 - F(x^*))}{\varepsilon^2}\sqrt{\frac{1+\alpha}{p}}L\right)$ iterations suffice. If $\theta \neq 0$ (for instance for the MARINA estimator equation 17), if we choose the batch-size $b = \Omega\left(\frac{\sum_{i=1}^n \sigma_i^2}{n^2\varepsilon^2}\right)$, then $2C\theta = \mathcal{O}(\varepsilon^2)$, so $\Omega\left(\frac{(\Psi_1 - F(x^*))}{\varepsilon^2}\sqrt{\frac{1+\alpha}{p}}L\right)$ iterations suffice.

If $F$ further satisfies the Lojasiewicz condition 7, we can obtain the following stronger result.

**Theorem B.2.** *Let $(g_k)_{k=0}^{K-1}$ be MARINA estimators. Assume that the gradient domination condition 7 holds. If the step-size satisfies*

$$0 < h \leq \min\left\{\frac{1}{14L}\sqrt{\frac{p}{1+\alpha}}, \frac{p}{6\mu}\right\}, \tag{30}$$

*we will have*

$$\mathrm{E}\left[F(x_K)\right] - F(x^*) \leq e^{-\mu Kh}\left(\Psi_2 - F(x^*)\right) + \frac{1 - e^{-K\mu h}}{\mu}C\theta, \tag{31}$$

*where $\Psi_2 = F(x_0) + \frac{1-e^{-\mu h}}{\mu}C\,\boldsymbol{G}_0, C = \frac{8L^2h^2\beta + 2\beta}{1-(1-p)(4L^2h^2\alpha+1)\beta}, \beta = e^{\mu h}$.*

Let $h = \min\{\frac{1}{14L}\sqrt{\frac{p}{1+\alpha}}, \frac{p}{6\mu}\}$ (hence $C = \mathcal{O}\left(\frac{1}{p}\right)$). To achieve $\mathrm{E}\left[F(x_K)\right] - F(x^*) \leq \varepsilon$ when $\theta = 0$ (for instance for the MARINA estimators equation 14 and equation 16), $\Omega(\max\{\sqrt{\frac{1+\alpha}{p}}\frac{L}{\mu}, \frac{1}{p}\}\log\left(\frac{\Psi_2 - F(x^*)}{\varepsilon}\right))$ iterations suffice. If $\theta \neq 0$ (for instance for the MARINA estimator equation 17), if we choose the batch-size $b = \Omega\left(\frac{\sum_{i=1}^n \sigma_i^2}{\mu n^2 \varepsilon}\right)$, then term $\frac{1-e^{-K\mu h}}{\mu}C\theta$ is of order $\mathcal{O}(\varepsilon)$, so $\Omega\left(\max\{\sqrt{\frac{1+\alpha}{p}}\frac{L}{\mu}, \frac{1}{p}\}\log(\frac{\Psi_2 - F(x^*)}{\varepsilon})\right)$ iterations suffice.

Since we recover the convergence rates of Gorbunov et al. (2021), we refer to the latter paper for a discussion of these results.

## C Proofs

### C.1 Proof of Proposition 3.3

For gradient estimator 14, we have

$$
\begin{aligned}
&\mathrm{E}\left[\left\|g_{k+1} - \nabla F(x_{k+1})\right\|^2 \mid x_{k+1}, x_k\right] \\
&= (1-p)\mathrm{E}\left[\left\|g_k + \frac{1}{n}\sum_{i=1}^n \mathcal{Q}(\nabla F_i(x_{k+1}) - \nabla F_i(x_k)) - \nabla F(x_{k+1})\right\|^2 \mid x_{k+1}, x_k\right] \\
&= (1-p)\mathrm{E}\left[\left\|\frac{1}{n}\sum_{i=1}^n \mathcal{Q}(\nabla F_i(x_{k+1}) - \nabla F_i(x_k)) - \nabla F(x_{k+1}) + \nabla F(x_k)\right\|^2 \mid x_{k+1}, x_k\right] \\
&\quad + (1-p)\mathrm{E}\left[\left\|g_k - \nabla F(x_k)\right\|^2 \mid x_k\right].
\end{aligned}
\tag{32}
$$

Since $\mathcal{Q}\left(\nabla F_1\left(x_{k+1}\right) - \nabla F_1\left(x_k\right)\right), \ldots, \mathcal{Q}\left(\nabla F_n\left(x_{k+1}\right) - \nabla F_n\left(x_k\right)\right)$ are independent random vectors, now we have

$$
\begin{aligned}
&\mathrm{E}\left[\left\|g_{k+1} - \nabla F(x_{k+1})\right\|^2 \mid x_{k+1}, x_k\right] \\
&= (1-p)\mathrm{E}\left[\left\|\frac{1}{n}\sum_{i=1}^n \left(\mathcal{Q}(\nabla F_i(x_{k+1}) - \nabla F_i(x_k)) - \nabla F_i(x_{k+1}) + \nabla F_i(x_k)\right)\right\|^2 \mid x_{k+1}, x_k\right] \\
&\quad + (1-p)\mathrm{E}\left[\left\|g_k - \nabla F(x_k)\right\|^2 \mid x_k\right] \\
&= \frac{1-p}{n^2}\sum_{i=1}^n \mathrm{E}\left[\left\|\mathcal{Q}(\nabla F_i(x_{k+1}) - \nabla F_i(x_k)) - \nabla F_i(x_{k+1}) + \nabla F_i(x_k)\right\|^2 \mid x_{k+1}, x_k\right] \\
&\quad + (1-p)\mathrm{E}\left[\left\|g_k - \nabla F(x_k)\right\|^2 \mid x_k\right] \\
&\leq \frac{(1-p)\omega}{n^2}\sum_{i=1}^n \mathrm{E}\left[\left\|\nabla F_i(x_{k+1}) - \nabla F_i(x_k)\right\|^2 \mid x_{k+1}, x_k\right] + (1-p)\mathrm{E}\left[\left\|g_k - \nabla F(x_k)\right\|^2 \mid x_k\right].
\end{aligned}
\tag{33}
$$

Use Equation (15) and the tower property, we obtain

$$
\begin{aligned}
&\mathrm{E}\left[\left\|g_{k+1} - \nabla F(x_{k+1})\right\|^2\right] \\
&= \mathrm{E}\left[\mathrm{E}\left[\left\|g_{k+1} - \nabla F(x_{k+1})\right\|^2 \mid x_{k+1}, x_k\right]\right] \\
&\leq \frac{(1-p)\omega}{n^2}\sum_{i=1}^n L_i^2 \mathrm{E}\left[\left\|x_{k+1} - x_k\right\|^2\right] + (1-p)\mathrm{E}\left[\left\|g_k - \nabla F(x_k)\right\|^2\right] \\
&= (1-p)L^2 \frac{\omega\sum_{i=1}^n L_i^2}{n^2 L^2}\mathrm{E}\left[\left\|x_{k+1} - x_k\right\|^2\right] + (1-p)\mathrm{E}\left[\left\|g_k - \nabla F(x_k)\right\|^2\right],
\end{aligned}
\tag{34}
$$

so $\alpha = \frac{\omega \sum_{i=1}^{n} L_i^2}{n^2 L^2}, \theta = 0$.

## C.2 Proof of Proposition 3.4

For gradient estimator 16 (finite sum case, that is for each $i \in [n]$, $F_i := \frac{1}{N} \sum_{j=1}^{N} F_{ij}$), we have

$$
\begin{aligned}
&\mathrm{E}\left[\|g_{k+1} - \nabla F(x_{k+1})\|^2 \mid x_{k+1}, x_k\right] \\
&= (1-p)\mathrm{E}\left[\left\|g_k + \frac{1}{n}\sum_{i=1}^{n} \mathcal{Q}\left(\frac{1}{b'}\sum_{j \in I'_{i,k}} (\nabla F_{ij}(x_{k+1}) - \nabla F_{ij}(x_k))\right) - \nabla F(x_{k+1})\right\|^2 \mid x_{k+1}, x_k\right] \\
&= (1-p)\mathrm{E}\left[\left\|\frac{1}{n}\sum_{i=1}^{n} \mathcal{Q}\left(\frac{1}{b'}\sum_{j \in I'_{i,k}} (\nabla F_{ij}(x_{k+1}) - \nabla F_{ij}(x_k))\right) - \nabla F(x_{k+1}) + \nabla F(x_k)\right\|^2 \mid x_{k+1}, x_k\right] \\
&\quad + (1-p)\mathrm{E}\left[\|g_k - \nabla F(x_k)\|^2 \mid x_k\right].
\end{aligned}
\tag{35}
$$

Next, we use the notation: $\widetilde{\Delta}_i^k = \frac{1}{b'}\sum_{j \in I'_{i,k}} (\nabla F_{ij}(x_{k+1}) - \nabla F_{ij}(x_k))$ and $\Delta_i^k = \nabla F_i(x_{k+1}) - \nabla F_i(x_k)$. They satisfy $\mathbb{E}\left[\widetilde{\Delta}_i^k \mid x_{k+1}, x_k\right] = \Delta_i^k$ for all $i \in [n]$. Moreover, $\mathcal{Q}\left(\tilde{\Delta}_1^k\right), \ldots, \mathcal{Q}\left(\tilde{\Delta}_n^k\right)$ are independent random vectors, now we have

$$
\begin{aligned}
&\mathrm{E}\left[\|g_{k+1} - \nabla F(x_{k+1})\|^2 \mid x_{k+1}, x_k\right] \\
&= (1-p)\mathrm{E}\left[\left\|\frac{1}{n}\sum_{i=1}^{n}\left(\mathcal{Q}(\widetilde{\Delta}_i^k) - \Delta_i^k\right)\right\|^2 \mid x_{k+1}, x_k\right] + (1-p)\mathrm{E}\left[\|g_k - \nabla F(x_k)\|^2 \mid x_k\right] \\
&= \frac{1-p}{n^2}\sum_{i=1}^{n}\mathrm{E}\left[\left\|\mathcal{Q}(\widetilde{\Delta}_i^k) - \widetilde{\Delta}_i^k + \widetilde{\Delta}_i^k - \Delta_i^k\right\|^2 \mid x_{k+1}, x_k\right] + (1-p)\mathrm{E}\left[\|g_k - \nabla F(x_k)\|^2 \mid x_k\right] \\
&= \frac{1-p}{n^2}\sum_{i=1}^{n}\left(\mathrm{E}\left[\left\|\mathcal{Q}(\widetilde{\Delta}_i^k) - \widetilde{\Delta}_i^k\right\|^2 \mid x_{k+1}, x_k\right] + \mathrm{E}\left[\left\|\widetilde{\Delta}_i^k - \Delta_i^k\right\|^2 \mid x_{k+1}, x_k\right]\right) + (1-p)\mathrm{E}\left[\|g_k - \nabla F(x_k)\|^2 \mid x_k\right] \\
&= \frac{1-p}{n^2}\sum_{i=1}^{n}\left(\omega\mathrm{E}\left[\left\|\widetilde{\Delta}_i^k\right\|^2 \mid x_{k+1}, x_k\right] + \mathrm{E}\left[\left\|\widetilde{\Delta}_i^k - \Delta_i^k\right\|^2 \mid x_{k+1}, x_k\right]\right) + (1-p)\mathrm{E}\left[\|g_k - \nabla F(x_k)\|^2 \mid x_k\right] \\
&= \frac{1-p}{n^2}\sum_{i=1}^{n}\left(\omega\mathrm{E}\left[\left\|\Delta_i^k\right\|^2 \mid x_{k+1}, x_k\right] + (1+\omega)\mathrm{E}\left[\left\|\widetilde{\Delta}_i^k - \Delta_i^k\right\|^2 \mid x_{k+1}, x_k\right]\right) + (1-p)\mathrm{E}\left[\|g_k - \nabla F(x_k)\|^2 \mid x_k\right].
\end{aligned}
\tag{36}
$$

Next we need to calculate $\mathrm{E}\left[\left\|\widetilde{\Delta}_i^k - \Delta_i^k\right\|^2 \mid x_{k+1}, x_k\right]$. For convenience, we will denote $a_{ij} := \nabla F_{ij}(x_{k+1}) - \nabla F_{ij}(x_k)$ and $a_i := \nabla F_i(x_{k+1}) - \nabla F_i(x_k)$. Define

$$
\chi_s = \begin{cases} 1 & \text{with prob.} \frac{1}{N} \\ 2 & \text{with prob.} \frac{1}{N} \\ \vdots & \\ N & \text{with prob.} \frac{1}{N} \end{cases},
\tag{37}
$$

$\{\chi_s\}_{s=1}^{b'}$ independent with each other. Let $I'_{i,k} = \bigcup_{s=1}^{b'} \chi_s$, so

$$\widetilde{\Delta}_i^k = \frac{1}{b'} \sum_{j \in I'_{i,k}} (\nabla F_{ij}(x_{k+1}) - \nabla F_{ij}(x_k))$$

$$= \frac{1}{b'} \sum_{s=1}^{b'} \sum_{j=1}^{N} 1_{\chi_s=j} a_{ij},$$

(38)

then

$$\mathrm{E}\left[\left\|\widetilde{\Delta}_i^k\right\|^2 - \Delta_i^k \mid x_{k+1}, x_k\right]$$

$$= \mathrm{E}\left[\left\|\frac{1}{b'} \sum_{s=1}^{b'} \sum_{j=1}^{N} 1_{\chi_s=j}(a_{ij} - a_i)\right\|^2 \mid x_{k+1}, x_k\right]$$

$$= \frac{1}{b'^2}\left(\sum_{s=1}^{b'} \mathrm{E}\left[\left\|\sum_{j=1}^{N} 1_{\chi_s=j}(a_{ij} - a_i)\right\|^2 \mid x_{k+1}, x_k\right] + \sum_{s \neq s'} \mathrm{E}\left[\left\langle \sum_{j=1}^{N} 1_{\chi_s=j}(a_{ij} - a_i), \sum_{j=1}^{N} 1_{\chi'_s=j}(a_{ij} - a_i) \right\rangle \mid x_{k+1}, x_k\right]\right)$$

$$= \frac{1}{b'^2}\left(\sum_{s=1}^{b'} \mathrm{E}\left[\left\|\sum_{j=1}^{N} 1_{\chi_s=j}(a_{ij} - a_i)\right\|^2 \mid x_{k+1}, x_k\right]\right.$$

$$\left. + \sum_{s \neq s'} \left\langle \mathrm{E}\left[\sum_{j=1}^{N} 1_{\chi_s=j}(a_{ij} - a_i) \mid x_{k+1}, x_k\right], \mathrm{E}\left[\sum_{j=1}^{N} 1_{\chi_{s'}=j}(a_{ij} - a_i) \mid x_{k+1}, x_k\right] \right\rangle\right)$$

$$= \frac{1}{b'}\left(\mathrm{E}\left[\left\|\sum_{j=1}^{N} 1_{\chi_1=j} a_{ij}\right\|^2 \mid x_{k+1}, x_k\right] - \|a_i\|^2\right)$$

$$\leq \frac{1}{b'}\mathrm{E}\left[\left\|\sum_{j=1}^{N} 1_{\chi_1=j} a_{ij}\right\|^2 \mid x_{k+1}, x_k\right]$$

$$= \frac{1}{b'}\left(\sum_{j=1}^{N} \mathrm{E}\left[\|1_{\chi_1=j}\|^2\right] \|a_{ij}\|^2 + \sum_{j \neq j'} \mathrm{E}\left[\langle 1_{\chi_1=j}, 1_{\chi_1=j'}\rangle\right] \langle a_{ij}, a_{ij'}\rangle\right)$$

$$= \frac{1}{b'} \frac{1}{N} \sum_{j=1}^{N} \|a_{ij}\|^2$$

$$\leq \frac{\mathcal{L}_i^2}{b'} \|x_{k+1} - x_k\|^2.$$

(39)

Use Equation (15) and the tower property, we get

$$\mathrm{E}\left[\|g_{k+1} - \nabla F(x_{k+1})\|^2\right]$$

$$= \mathrm{E}\left[\mathrm{E}\left[\|g_{k+1} - \nabla F(x_{k+1})\|^2 \mid x_{k+1}, x_k\right]\right]$$

$$\leq \frac{1-p}{n^2} \sum_{i=1}^{n} \left(\omega L_i^2 + \frac{(1+\omega)\mathcal{L}_i^2}{b'}\right) \mathrm{E}\left[\|x_{k-1} - x_k\|^2\right] + (1-p)\mathrm{E}\left[\|g_k - \nabla F(x_k)\|^2\right]$$

(40)

$$\leq (1-p)L^2 \frac{\omega \sum_{i=1}^{n} L_i^2 + (1+\omega)\frac{\sum_{i=1}^{n} \mathcal{L}_i^2}{b'}}{n^2 L^2} \mathrm{E}\left[\|x_{k-1} - x_k\|^2\right] + (1-p)\mathrm{E}\left[\|g_k - \nabla F(x_k)\|^2\right],$$

so $\alpha = \dfrac{\omega \sum_{i=1}^{n} L_i^2 + (1+\omega)\frac{\sum_{i=1}^{n} \mathcal{L}_i^2}{b'}}{n^2 L^2}, \theta = 0.$

## C.3    Proof of Theorem 3.5

For gradient estimator 17 (online case, that is for each $i \in [n]$, $F_i := \mathbb{E}_{\xi_i \sim \mathcal{D}_i}[F_{\xi_i}]$), we first have

$$
\begin{aligned}
&\mathrm{E}\left[\left[\|g_{k+1} - \nabla F(x_{k+1})\|^2 \mid x_{k+1}, x_k\right]\right] \\
&= (1-p)\mathrm{E}\left[\left[\left\|g_k + \frac{1}{n}\sum_{i=1}^{n}\mathcal{Q}\left(\frac{1}{b'}\sum_{\xi \in I'_{i,k}}(\nabla F_\xi(x_{k+1}) - \nabla F_\xi(x_k))\right) - \nabla F(x_{k+1})\right\|^2 \mid x_{k+1}, x_k\right]\right] \\
&\quad + \frac{p}{n^2 b^2}\mathrm{E}\left[\left[\left\|\sum_{i=1}^{n}\sum_{\xi \in I_{i,k}}(\nabla F_\xi(x_{k+1}) - \nabla F(x_{k+1}))\right\|^2 \mid x_{k+1}\right]\right] \\
&= (1-p)\mathrm{E}\left[\left[\left\|\frac{1}{n}\sum_{i=1}^{n}\mathcal{Q}\left(\frac{1}{b'}\sum_{\xi \in I'_{i,k}}(\nabla F_\xi(x_{k+1}) - \nabla F_\xi(x_k))\right) - \nabla F(x_{k+1}) + \nabla F(x_k)\right\|^2 \mid x_{k+1}, x_k\right]\right] \\
&\quad + (1-p)\mathrm{E}\left[\left[\|g_k - \nabla F(x_k)\|^2 \mid x_k\right]\right] + \frac{p}{n^2 b^2}\sum_{i=1}^{n}\sum_{\xi \in I_{i,k}}\mathrm{E}\left[\left[\|\nabla F_\xi(x_{k+1}) - \nabla F(x_{k+1})\|^2 \mid x_{k+1}\right]\right] \\
&\leq (1-p)\mathrm{E}\left[\left[\left\|\frac{1}{n}\sum_{i=1}^{n}\mathcal{Q}\left(\frac{1}{b'}\sum_{\xi \in I'_{i,k}}(\nabla F_\xi(x_{k+1}) - \nabla F_\xi(x_k))\right) - \nabla F(x_{k+1}) + \nabla F(x_k)\right\|^2 \mid x_{k+1}, x_k\right]\right] \\
&\quad + (1-p)\mathrm{E}\left[\left[\|g_k - \nabla F(x_k)\|^2 \mid x_k\right]\right] + \frac{p\sum_{i=1}^{n}\sigma_i^2}{n^2 b},
\end{aligned}
$$
$$(41)$$

here $I'_{i,k}$ consists of $b'$ elements i.i.d. sampled from distribution $\mathcal{D}_i$. In the following, we use the notation: $\widetilde{\Delta}_i^k = \frac{1}{b'}\sum_{\xi \in I'_{i,k}}(\nabla F_\xi(x_{k+1}) - \nabla F_\xi(x_k))$ and $\Delta_i^k = \nabla F_i(x_{k+1}) - \nabla F_i(x_k)$. They satisfy $\mathbb{E}\left[\widetilde{\Delta}_i^k \mid x_{k+1}, x_k\right] =$

$\Delta_i^k$ for all $i \in [n]$. Moreover, $\mathcal{Q}\left(\tilde{\Delta}_1^k\right), \ldots, \mathcal{Q}\left(\tilde{\Delta}_n^k\right)$ are independent random vectors, then we have

$$
\mathrm{E}\left[\|g_{k+1} - \nabla F(x_{k+1})\|^2 \mid x_{k+1}, x_k\right]
$$

$$
\leq (1-p)\mathrm{E}\left[\left\|\frac{1}{n}\sum_{i=1}^n \left(\mathcal{Q}(\tilde{\Delta}_i^k) - \Delta_i^k\right)\right\|^2 \mid x_{k+1}, x_k\right] + (1-p)\mathrm{E}\left[\|g_k - \nabla F(x_k)\|^2 \mid x_k\right] + \frac{p\sum_{i=1}^n \sigma_i^2}{n^2 b}
$$

$$
= \frac{1-p}{n^2}\sum_{i=1}^n \mathrm{E}\left[\left\|\mathcal{Q}(\tilde{\Delta}_i^k) - \tilde{\Delta}_i^k + \tilde{\Delta}_i^k - \Delta_i^k\right\|^2 \mid x_{k+1}, x_k\right] + (1-p)\mathrm{E}\left[\|g_k - \nabla F(x_k)\|^2 \mid x_k\right] + \frac{p\sum_{i=1}^n \sigma_i^2}{n^2 b}
$$

$$
= \frac{1-p}{n^2}\sum_{i=1}^n \left(\mathrm{E}\left[\left\|\mathcal{Q}(\tilde{\Delta}_i^k) - \tilde{\Delta}_i^k\right\|^2 \mid x_{k+1}, x_k\right] + \mathrm{E}\left[\left\|\tilde{\Delta}_i^k - \Delta_i^k\right\|^2 \mid x_{k+1}, x_k\right]\right) + (1-p)\mathrm{E}\left[\|g_k - \nabla F(x_k)\|^2 \mid x_k\right]
$$

$$
+ \frac{p\sum_{i=1}^n \sigma_i^2}{n^2 b}
$$

$$
= \frac{1-p}{n^2}\sum_{i=1}^n \left(\omega\mathrm{E}\left[\left\|\tilde{\Delta}_i^k\right\|^2 \mid x_{k+1}, x_k\right] + \mathrm{E}\left[\left\|\tilde{\Delta}_i^k - \Delta_i^k\right\|^2 \mid x_{k+1}, x_k\right]\right) + (1-p)\mathrm{E}\left[\|g_k - \nabla F(x_k)\|^2 \mid x_k\right]
$$

$$
+ \frac{p\sum_{i=1}^n \sigma_i^2}{n^2 b}
$$

$$
= \frac{1-p}{n^2}\sum_{i=1}^n \left(\omega\mathrm{E}\left[\left\|\Delta_i^k\right\|^2 \mid x_{k+1}, x_k\right] + (1+\omega)\mathrm{E}\left[\left\|\tilde{\Delta}_i^k - \Delta_i^k\right\|^2 \mid x_{k+1}, x_k\right]\right) + (1-p)\mathrm{E}\left[\|g_k - \nabla F(x_k)\|^2 \mid x_k\right]
$$

$$
+ \frac{p\sum_{i=1}^n \sigma_i^2}{n^2 b}.
$$

(42)

Now we need to calculate $\mathrm{E}\left[\left\|\tilde{\Delta}_i^k - \Delta_i^k\right\|^2 \mid x_{k+1}, x_k\right]$. Let $\xi_{i,s}^k \sim \mathcal{D}_i, s = 1, 2, \ldots, b'$ be $b'$ i.i.d. random variables, $I'_{i,k} := \bigcup_{s=1}^{b'} \xi_{i,s}^k$ and denote $a_{\xi_{i,s}^k} := \nabla F_{\xi_{i,s}^k}(x_{k+1}) - \nabla F_{\xi_{i,s}^k}(x_k), a_i := \nabla F_i(x_{k+1}) - \nabla F_i(x_k)$, then

$$
\tilde{\Delta}_i^k = \frac{1}{b'}\sum_{\xi \in I'_{i,k}} a_\xi = \frac{1}{b'}\sum_{k=1}^{b'} a_{\xi_{i,s}^k}.
$$

(43)

So

$$
\mathrm{E}\left[\left\|\widetilde{\Delta}_i^k - \Delta_i^k\right\|^2 \mid x_{k+1}, x_k\right]
$$

$$
= \mathrm{E}\left[\left\|\frac{1}{b'}\sum_{k=1}^{b'}\left(a_{\xi_{i,s}^k} - a_i\right)\right\|^2 \mid x_{k+1}, x_k\right]
$$

$$
= \frac{1}{b'^2}\sum_{k=1}^{b'}\mathrm{E}\left[\left\|a_{\xi_{i,s}^k} - a_i\right\|^2 \mid x_{k+1}, x_k\right] + \frac{1}{b'^2}\sum_{k\neq k'}\mathrm{E}\left[\left\langle a_{\xi_{i,s}^k} - a_i, a_{\chi_{i,k'}} - a_i\right\rangle \mid x_{k+1}, x_k\right]
$$

$$
= \frac{1}{b'^2}\sum_{k=1}^{b'}\mathrm{E}\left[\left\|a_{\xi_{i,s}^k} - a_i\right\|^2 \mid x_{k+1}, x_k\right] \tag{44}
$$

$$
= \frac{1}{b'^2}\sum_{k=1}^{b'}\mathrm{E}\left[\left\|a_{\xi_{i,s}^k} - a_i\right\|^2 \mid x_{k+1}, x_k\right]
$$

$$
= \frac{1}{b'}\left(\mathrm{E}\left[\left\|a_{\xi_{i,1}^k}\right\|^2 \mid x_{k+1}, x_k\right] - \|a_i\|^2\right)
$$

$$
\leq \frac{1}{b'}\mathrm{E}\left[\left\|a_{\xi_{i,1}^k}\right\|^2 \mid x_{k+1}, x_k\right]
$$

$$
\leq \frac{\mathcal{L}_i^2}{b'}\|x_{k+1} - x_k\|^2.
$$

Combine with the tower property, we finally get

$$
\mathrm{E}\left[\|g_{k+1} - \nabla F(x_{k+1})\|^2\right]
$$

$$
= \mathrm{E}\left[\mathrm{E}\left[\|g_{k+1} - \nabla F(x_{k+1})\|^2 \mid x_{k+1}, x_k\right]\right]
$$

$$
= \frac{1-p}{n^2}\sum_{i=1}^{n}\left(\omega L_i^2 + \frac{(1+\omega)\mathcal{L}_i^2}{b'}\right)\mathrm{E}\left[\|x_{k+1} - x_k\|^2\right] + (1-p)\mathrm{E}\left[\|g_k - \nabla F(x_k)\|^2\right] + \frac{p\sum_{i=1}^{n}\sigma_i^2}{n^2 b}
$$

$$
= (1-p)L^2\frac{\omega\sum_{i=1}^{n}L_i^2 + (1+\omega)\frac{\sum_{i=1}^{n}\mathcal{L}_i^2}{b'}}{n^2 L^2}\mathrm{E}\left[\|x_{k+1} - x_k\|^2\right] + (1-p)\mathrm{E}\left[\|g_k - \nabla F(x_k)\|^2\right] + \frac{p\sum_{i=1}^{n}\sigma_i^2}{n^2 b}, \tag{45}
$$

so $\alpha = \frac{\omega\sum_{i=1}^{n}L_i^2 + (1+\omega)\frac{\sum_{i=1}^{n}\mathcal{L}_i^2}{b'}}{n^2 L^2}, \theta = \frac{p\sum_{i=1}^{n}\sigma_i^2}{n^2 b}$.

### C.4  Proof of Theorem 4.1

The following lemma is an integral form of Grönwall inequality from Amann (2011, Chapter II.), which plays an important role in the proof of Theorem 4.1 and B.2.

**Lemma C.1** (Grönwall Inequality). *Assume $\phi, B : [0, T] \to \mathbb{R}$ are bounded non-negative measurable function and $C : [0, T] \to \mathbb{R}$ is a non-negative integrable function with the property that*

$$
\phi(t) \leq B(t) + \int_0^t C(\tau)\phi(\tau)d\tau \quad \text{for all } t \in [0, T] \tag{46}
$$

*Then*

$$
\phi(t) \leq B(t) + \int_0^t B(s)C(s)\exp\left(\int_s^t C(\tau)d\tau\right)ds \quad \text{for all } t \in [0, T]. \tag{47}
$$

We also need the following lemma from Chewi et al. (2021, Lemma 16.).

**Lemma C.2.** *Assume that $\nabla F$ is $L$-Lipschitz. For any probability measure $\mu$, it holds that*

$$
\mathbb{E}_\mu\left[\|\nabla F\|^2\right] \leq \mathbb{E}_\mu\left[\left\|\nabla\log(\frac{\mu}{\pi})\right\|^2\right] + 2dL = J_\pi(\mu) + 2dL. \tag{48}
$$

Remind you the definition of KL divergence and Fisher information:

$$D_{\mathrm{KL}}\left(\rho_t \mid \pi\right) := \int_{\mathbb{R}^d} \log(\frac{\rho_t}{\pi})(x)d\rho_t, \quad J_\pi\left(\rho_t\right) := \int_{\mathbb{R}^d} \left\|\nabla \log(\frac{\rho_t}{\pi})\right\|^2 d\rho_t. \tag{49}$$

We follow the proof of Vempala & Wibisono (2019, Lemma 3). Consider the following SDE

$$dx_t = -f_\xi(x_0)dt + \sqrt{2}dB_t, \tag{50}$$

where $f_.(\cdot) : \mathbb{R}^d \times \Xi \to \mathbb{R}^d$, $\Xi$ is of some probability space $(\Xi, \rho, \mathcal{F})$, let $\rho_{0t}(x_0, \xi, x_t)$ denote the joint distribution of $(x_0, \xi, x_t)$, which we write in terms of the conditionals and marginals as

$$\rho_{0t}\left(x_0, \xi, x_t\right) = \rho_0\left(x_0, \xi\right)\rho_{t|0}\left(x_t \mid x_0, \xi\right) = \rho_t\left(x_t\right)\rho_{0|t}\left(x_0, \xi \mid x_t\right).$$

Conditioning on $(x_0, \xi)$, the drift vector field $f_\xi(x_0)$ is a constant, so the Fokker-Planck formula for the conditional density $\rho_{t|0}\left(x_t \mid x_0\right)$ is

$$\frac{\partial \rho_{t|0}\left(x_t \mid x_0, \xi\right)}{\partial t} = \nabla \cdot \left(\rho_{t|0}\left(x_t \mid x_0, \xi\right) f_\xi\left(x_0\right)\right) + \Delta\rho_{t|0}\left(x_t \mid x_0, \xi\right)$$

To derive the evolution of $\rho_t$, we take expectation over $(x_0, \xi) \sim \rho_0$, we obtain

$$\begin{aligned}
\frac{\partial \rho_t(x)}{\partial t} &= \int_{\mathbb{R}^d \times \Xi} \frac{\partial \rho_{t|0}\left(x \mid x_0, \xi\right)}{\partial t}\rho_0\left(x_0, \xi\right)dx_0 d\xi \\
&= \int_{\mathbb{R}^d \times \Xi} \left(\nabla \cdot \left(\rho_{t|0}\left(x_t \mid x_0, \xi\right) f_\xi\left(x_0\right)\right) + \Delta\rho_{t|0}\left(x_t \mid x_0, \xi\right)\right)\rho_0\left(x_0, \xi\right)dx_0 d\xi \\
&= \int_{\mathbb{R}^d \times \Xi} \left(\nabla \cdot \left(\rho_{0t}\left(x, x_0, \xi\right) f_\xi\left(x_0\right)\right) + \Delta\rho_{0t}\left(x, x_0, \xi\right)\right)dx_0 d\xi \\
&= \nabla \cdot \left(\rho_t(x)\int_{\mathbb{R}^d \times \Xi}\rho_{0|t}\left(x_0, \xi \mid x\right) f_\xi\left(x_0\right)dx_0 d\xi\right) + \Delta\rho_t(x) \\
&= \nabla \cdot \left(\rho_t(x)\mathbb{E}_{\rho_{0|t}}\left[f_\xi\left(x_0\right) \mid x_t = x\right]\right) + \Delta\rho_t(x),
\end{aligned} \tag{51}$$

so we have

$$\begin{aligned}
\frac{dD_{\mathrm{KL}}\left(\rho_t \mid \pi\right)}{dt} &= \int_{\mathbb{R}^d} \left(\nabla \cdot \left(\rho_t(x)\mathbb{E}_{\rho_{0|t}}\left[f_\xi\left(x_0\right) \mid x_t = x\right]\right) + \Delta\rho_t(x)\right)\log(\frac{\rho_t}{\pi})(x)dx \\
&= -\int_{\mathbb{R}^d} \left\langle\mathbb{E}_{\rho_{0|t}}\left[f_\xi\left(x_0\right) \mid x_t = x\right] + \nabla\log(\rho_t)(x), \nabla\log(\frac{\rho_t}{\pi})(x)\right\rangle\rho_t(x)dx \\
&= -\int_{\mathbb{R}^d} \left\langle\nabla\log(\frac{\rho_t}{\pi})(x) - \nabla\log(\frac{\rho_t}{\pi})(x) + \mathbb{E}_{\rho_{0|t}}\left[f_\xi\left(x_0\right) \mid x_t = x\right] + \nabla\log(\rho_t)(x), \nabla\log(\frac{\rho_t}{\pi})(x)\right\rangle\rho_t(x)dx \\
&= -\int_{\mathbb{R}^d} \left\langle\nabla\log(\frac{\rho_t}{\pi})(x) + \mathbb{E}_{\rho_{0|t}}\left[f_\xi\left(x_0\right) \mid x_t = x\right] - \nabla F(x), \nabla\log(\frac{\rho_t}{\pi})(x)\right\rangle\rho_t(x)dx \\
&= -J_\pi\left(\rho_t\right) - \int_{\mathbb{R}^d} \left\langle\mathbb{E}_{\rho_{0|t}}\left[f_\xi\left(x_0\right) \mid x_t = x\right] - \nabla F(x), \nabla\log(\frac{\rho_t}{\pi})(x)\right\rangle\rho_t(x)dx \\
&\leq -J_\pi\left(\rho_t\right) + \frac{1}{4}J_\pi\left(\rho_t\right) + \int_{\mathbb{R}^d} \left\langle\mathbb{E}_{\rho_{0|t}}\left[f_\xi\left(x_0\right) \mid x_t = x\right] - \nabla F(x), \mathbb{E}_{\rho_{0|t}}\left[f_\xi\left(x_0\right) \mid x_t = x\right] - \nabla F(x)\right\rangle\rho_t(x)dx \\
&\leq -\frac{3}{4}J_\pi\left(\rho_t\right) + \mathrm{E}\left[\left\|\mathbb{E}\left[f_\xi(x_0) - \nabla F(x_t) \mid x_t\right]\right\|^2\right] \\
&\leq -\frac{3}{4}J_\pi\left(\rho_t\right) + \mathrm{E}\left[\mathbb{E}\left[\left\|f_\xi(x_0) - \nabla F(x_t)\right\|^2 \mid x_t\right]\right] \\
&= -\frac{3}{4}J_\pi\left(\rho_t\right) + \mathrm{E}\left[\left\|f_\xi(x_0) - \nabla F(x_t)\right\|^2\right].
\end{aligned} \tag{52}$$

If we replace $f_\xi(x_0)$ by $g_k(x_k)$ in equation 52, we will have

$$
\begin{aligned}
\frac{dD_{\mathrm{KL}}\left(\rho_t \mid \pi\right)}{dt} &\leq -\frac{3}{4} J_\pi\left(\rho_t\right) + \mathrm{E}\left[\left\|\nabla F(x_t) - g_k\right\|^2\right] \\
&\leq -\frac{3}{4} J_\pi\left(\rho_t\right) + 2\mathrm{E}\left[\left\|\nabla F(x_t) - \nabla F(x_k)\right\|^2\right] + 2\mathrm{E}\left[\left\|\nabla F(x_k) - g_k\right\|^2\right] \\
&= -\frac{3}{4} J_\pi\left(\rho_t\right) + 2\underbrace{\mathrm{E}\left[\left\|\nabla F(x_t) - \nabla F(x_k)\right\|^2\right]}_{A} + 2\mathbf{G}_k.
\end{aligned}
\tag{53}
$$

We bound term $A$, denote $\mathcal{F}_t^k$ the filtration generated by $\{B_s\}_{s=0}^{kh+t}$, then

$$
A = \mathbb{E}_{\rho_t}\left[\mathbb{E}\left[\left\|\nabla F(x_t) - \nabla F(x_k)\right\|^2 \mid \mathcal{F}_t^k\right]\right],
\tag{54}
$$

next we estimate the inner expectation,

$$
\begin{aligned}
\mathbb{E}\left[\left\|\nabla F(x_t) - \nabla F(x_k)\right\|^2 \mid \mathcal{F}_t^k\right] &\leq L^2 \mathbb{E}\left[\left\|x_t - x_k\right\|^2 \mid \mathcal{F}_t^k\right] \\
&= L^2 \mathbb{E}\left[\left\|t g_k + \sqrt{2}\left(B_{kh+t} - B_{kh}\right)\right\|^2 \mid \mathcal{F}_t^k\right] \\
&= L^2 t^2 \left\|g_k\right\|^2 + 2L^2 dt \\
&\leq L^2 h^2 \left\|g_k\right\|^2 + 2L^2 dh \\
&= L^2 \mathbb{E}\left[\left\|x_{k+1} - x_k\right\|^2 \mid \mathcal{F}_h^k\right],
\end{aligned}
\tag{55}
$$

from equation 55 and equation 53, we finally have

$$
\frac{dD_{\mathrm{KL}}\left(\rho_t \mid \pi\right)}{dt} \leq -\frac{3}{4} J_\pi\left(\rho_t\right) + 2L^2 \mathrm{E}\left[\left\|x_{k+1} - x_k\right\|^2\right] + 2\mathbf{G}_k.
\tag{56}
$$

Next, we use Lemma C.2 to bound $\mathrm{E}\left[\left\|x_{k+1} - x_k\right\|^2\right]$,

$$
\begin{aligned}
\mathrm{E}\left[\left\|x_{k+1} - x_k\right\|^2\right] &= h^2 \mathrm{E}\left[\left\|g_k\right\|^2\right] + 2dh \\
&\leq 2h^2 \left(\mathrm{E}\left[\left\|\nabla F(x_k) - g_k\right\|^2\right] + \mathrm{E}\left[\left\|\nabla F(x_k)\right\|^2\right]\right) + 2dh \\
&= 2h^2 \mathrm{E}\left[\left\|\nabla F(x_k)\right\|^2\right] + 2h^2 \mathbf{G}_k + 2dh \\
&\leq 4h^2 \left(\mathrm{E}\left[\left\|\nabla F(x_t)\right\|\right] + \mathrm{E}\left[\left\|\nabla F(x_t) - \nabla F(x_k)\right\|^2\right]\right) + 2h^2 \mathbf{G}_k + 2dh \\
&\leq 4h^2 \mathrm{E}\left[\left\|\nabla F(x_t)\right\|\right] + 4L^2 h^2 \mathrm{E}\left[\left\|x_{k+1} - x_k\right\|^2\right] + 2h^2 \mathbf{G}_k + 2dh,
\end{aligned}
\tag{57}
$$

so let $h \leq \frac{1}{2\sqrt{2}L}$, we have

$$
\mathrm{E}\left[\left\|x_{k+1} - x_k\right\|^2\right] \leq 8h^2 \mathrm{E}\left[\left\|\nabla F(x_t)\right\|^2\right] + 4h^2 \mathbf{G}_k + 4dh.
\tag{58}
$$

Add $C\mathbf{G}_{k+1}$ to both sides of inequality equation 56, $C$ is some constant to be determined later, then use Theorem 3.1, the right hand side of equation 56 will be

$$
\begin{aligned}
RHS =&\; -\frac{3}{4}J_\pi\left(\rho_t\right) + 2L^2\mathrm{E}\left[\|x_{k+1}-x_k\|^2\right] + 2\mathbf{G}_k + C\mathbf{G}_{k+1}\\
\leq&\; -\frac{3}{4}J_\pi\left(\rho_t\right) + 2L^2\mathrm{E}\left[\|x_{k+1}-x_k\|^2\right] + 2\mathbf{G}_k + C\left((1-p)\mathbf{G}_k + (1-p)L^2\alpha\mathrm{E}\left[\|x_{k+1}-x_k\|^2\right] + \theta\right)\\
=&\; -\frac{3}{4}J_\pi\left(\rho_t\right) + \left(2L^2 + C(1-p)L^2\alpha\right)\mathrm{E}\left[\|x_{k+1}-x_k\|^2\right] + (2+C(1-p))\,\mathbf{G}_k + C\theta\\
\overset{equation\ 58}{\leq}&\; -\frac{3}{4}J_\pi\left(\rho_t\right) + \left(2L^2 + C(1-p)L^2\alpha\right)\left(8h^2\mathrm{E}\left[\|\nabla F(x_t)\|^2\right] + 4h^2\mathbf{G}_k + 4dh\right)\\
&\; + (2+C(1-p))\,\mathbf{G}_k + C\theta\\
\overset{Lemma\ C.2}{\leq}&\; -\frac{3}{4}J_\pi\left(\rho_t\right) + \left(2L^2 + C(1-p)L^2\alpha\right)\left(8h^2\left(J_\pi\left(\rho_t\right) + 2nL\right) + 4h^2\mathbf{G}_k + 4dh\right)\\
&\; + (2+C(1-p))\,\mathbf{G}_k + C\theta\\
=&\; -\left(\frac{3}{4} - 8h^2\left(2L^2 + C(1-p)L^2\alpha\right)\right)J_\pi\left(\rho_t\right) + \left(8L^2h^2 + C(1-p)\left(4L^2h^2\alpha+1\right) + 2\right)\mathbf{G}_k\\
&\; + \underbrace{\left(2L^2 + C(1-p)L^2\alpha\right)\left(8Lh^2d + 4dh\right) + C\theta}_{denote\ as\ \tau}\\
=&\; -\left(\frac{3}{4} - 8h^2\left(2L^2 + C(1-p)L^2\alpha\right)\right)J_\pi\left(\rho_t\right) + \left(8L^2h^2 + C(1-p)\left(4L^2h^2\alpha+1\right) + 2\right)\mathbf{G}_k + \tau.
\end{aligned}
$$
(59)

Choose parameter $\beta$ ($\beta = 1$ or $\beta = e^{\mu h}$) and let $C = \left(8L^2h^2 + C(1-p)\left(4L^2h^2\alpha+1\right) + 2\right)\beta$, solve this, we get

$$
C = \frac{8L^2h^2\beta + 2\beta}{1 - (1-p)\left(4L^2h^2\alpha+1\right)\beta}.
$$
(60)

To make sure $C \geq 0$, we should require $h \leq \frac{1}{2L}\sqrt{\frac{p}{(1-p)\alpha}}$ when $\beta = 1$, the case $\beta = e^{\mu h}$ is a bit complicated: when $h$ small (for example $h \leq \frac{1}{\mu}$). we have $\beta = e^{\mu h} \leq 1 + 2\mu h$, insert this into the denominator of $C$ and make sure it positive, that is

$$
1 - (1-p)\left(4L^2\alpha h^2 + 1\right)(1+2\mu h) > 0,
$$
(61)

which is equivalent to

$$
\underbrace{\frac{1-p}{p}8L^2\mu\alpha h^3}_{I} + \underbrace{\frac{1-p}{p}4L^2\alpha h^2}_{II} + \underbrace{\frac{1-p}{p}2\mu h}_{III} < 1,
$$
(62)

one simple solution for equation 62 is to let $I < \frac{1}{3}, II < \frac{1}{3}, III < \frac{1}{3}$, which is

$$
h < \min\{(\frac{p}{24L^2\mu\alpha(1-p)})^{1/3}, (\frac{p}{12L^2\alpha(1-p)})^{1/2}, \frac{p}{6\mu(1-p)}\}.
$$
(63)

Insert equation 60 into the parameter before $\mathrm{E}\left[\|\nabla F(x_t)\|^2\right]$ and require

$$
8h^2\left(2L^2 + C(1-p)L^2\alpha\right) = 8h^2\left(2L^2 + \frac{8L^2h^2\beta + 2\beta}{1 - (1-p)\left(4L^2h^2\alpha+1\right)\beta}(1-p)L^2\alpha\right) \leq \frac{1}{4},
$$
(64)

solve this we get

$$
h \leq \frac{1}{2L}\sqrt{\frac{1-(1-p)\beta}{16 + (1-p)(17\alpha-16)\beta}}.
$$
(65)

If $\beta = 1$, we need

$$
h \leq \frac{1}{10L}\sqrt{\frac{p}{1+\alpha}} \leq \frac{1}{2L}\sqrt{\frac{p}{17\alpha(1-p)}} \leq \frac{1}{2L}\min\{\sqrt{\frac{p}{16p + 17\alpha(1-p)}}, \sqrt{\frac{p}{\alpha(1-p)}}\}
$$
(66)

to guarantee $C \geq 0$ and equation 65.

The $\beta = e^{\mu h}$ case is complicated: from equation 63, we know $h < \frac{p}{6\mu(1-p)}$, so $\beta = e^{\mu h} \leq 1 + 2\mu h \leq 1 + \frac{p}{3(1-p)}$, insert this upper bound of $\beta$ into equation 65, we get a lower bound of the right hand side of equation 65, that is

$$
\frac{1}{2L} \sqrt{\frac{2p}{17\alpha(3 - 2p) + 32p}} = \frac{1}{2L} \sqrt{\frac{1 - (1 - p)(1 + \frac{p}{3(1-p)})}{16 + (1 - p)(17\alpha - 16)(1 + \frac{p}{3(1-p)})}}
$$
$$
\leq \frac{1}{2L} \sqrt{\frac{1 - (1 - p)\beta}{16 + (1 - p)(17\alpha - 16)\beta}}, \tag{67}
$$

So we need

$$
h < \min\{\frac{1}{2L} \sqrt{\frac{2p}{17\alpha(3 - 2p) + 32p}}, (\frac{p}{24L^2\mu\alpha(1 - p)})^{1/3}, (\frac{p}{12L^2\alpha(1 - p)})^{1/2}, \frac{p}{6\mu(1 - p)}\}. \tag{68}
$$

We can further simplify equation 68: the first and third term in equation 68 will be greater than $\underbrace{\frac{1}{14L} \sqrt{\frac{p}{1 + \alpha}}}_{a}$,

the fourth term is greater than $\underbrace{\frac{p}{6\mu}}_{b}$ and $\min\{a, b\}$ is a lower bound of the second term in equation 68, since

$\min\{a, b\} \leq a^{2/3}b^{1/3} = (\frac{p^2}{1176L^2\mu(1+\alpha)})^{1/3} \leq (\frac{p}{24L^2\mu\alpha(1-p)})^{1/3}$.

So finally when $\beta = e^{\mu h}$, we need

$$
h \leq \min\{\frac{1}{14L} \sqrt{\frac{p}{1 + \alpha}}, \frac{p}{6\mu}\} \tag{69}
$$

to guarantee $C \geq 0$ and equation 65.

Once we have $C \geq 0$ and equation 64, then

$$
\frac{dD_{\mathrm{KL}}(\rho_t \mid \pi)}{dt} + C\mathbf{G}_{k+1} \leq -\frac{1}{2}J_\pi(\rho_t) + \beta^{-1}C\mathbf{G}_k + C\tau. \tag{70}
$$

**Case I.** $\beta = 1$, integrate equation 70, we have then

$$
\int_{kh}^{(k+1)h} J_\pi(\rho_t) \, dt \leq 2\left(D_{\mathrm{KL}}(\rho_{kh} \mid \pi) + C\mathbf{G}_k h - \left(D_{\mathrm{KL}}(\rho_{(k+1)h} \mid \pi) + C\mathbf{G}_{k+1}h\right)\right) + C\tau h, \tag{71}
$$

use equation 71 for $k = 0, 1, 2, \cdots, K$, we finally have

$$
\frac{1}{Kh} \int_0^{Kh} J_\pi(\rho_t) \, dt \leq \frac{2\left(D_{\mathrm{KL}}(\rho_0 \mid \pi) + C\mathbf{G}_0 h - (D_{\mathrm{KL}}(\rho_{Kh} \mid \pi) + C\mathbf{G}_K h)\right)}{Kh} + \tau
$$
$$
\leq \frac{2(D_{\mathrm{KL}}(\rho_0 \mid \pi) + hC\mathbf{G}_0)}{Kh} + \tau. \tag{72}
$$

**Case II.** Suppose $\pi$ satisfies LSI with parameter $\mu$, that is

$$
D_{\mathrm{KL}}(\nu \mid \pi) \leq \frac{1}{2\mu}J_\pi(\nu), \tag{73}
$$

so with LSI, we have from equation 70 that

$$
\frac{dD_{\mathrm{KL}}(\rho_t \mid \pi)}{dt} + C\mathbf{G}_{k+1} \leq -\mu D_{\mathrm{KL}}(\rho_t \mid \pi) + \beta^{-1}C\mathbf{G}_k + \tau. \tag{74}
$$

Change equation 74 into its equivalent integral form, then it satisfies equation 46 with $\phi(t) = D_{\mathrm{KL}}(\rho_t \mid \pi)$, $B(t) = \left(\beta^{-1} C\mathbf{G}_k - C\mathbf{G}_{k+1} + \tau\right) t + D_{\mathrm{KL}}(\rho_{kh} \mid \pi)$, $C(t) = -\mu$, then by equation 47, we have

$$D_{\mathrm{KL}}(\rho_t \mid \pi) \leq e^{\mu t} D_{\mathrm{KL}}(\rho_{kh} \mid \pi) + \frac{1 - e^{-\mu t}}{\mu}\left(\beta^{-1} C\mathbf{G}_k - C\mathbf{G}_{k+1} + \tau\right), \tag{75}$$

let $t = h$ and $\beta = e^{\mu h}$, then we have

$$D_{\mathrm{KL}}\left(\rho_{(k+1)h} \mid \pi\right) + \frac{1 - e^{-\mu h}}{\mu} C\mathbf{G}_{k+1} \leq e^{-\mu h}\left(D_{\mathrm{KL}}(\rho_{kh} \mid \pi) + e^{\mu h}\frac{1 - e^{-\mu h}}{\mu}\beta^{-1} C\mathbf{G}_k\right) + \frac{1 - e^{-\mu h}}{\mu}\tau$$
$$= e^{-\mu h}\left(D_{\mathrm{KL}}(\rho_{kh} \mid \pi) + \frac{1 - e^{-\mu h}}{\mu} C\mathbf{G}_k\right) + \frac{1 - e^{-\mu h}}{\mu}\tau, \tag{76}$$

use equation 76 for $k = 0, 1, 2, \cdots, K-1$, we have finally

$$D_{\mathrm{KL}}(\rho_{Kh} \mid \pi) \leq D_{\mathrm{KL}}(\rho_{Kh} \mid \pi) + \frac{1 - e^{-\mu h}}{\mu} C\mathbf{G}_K$$
$$\leq e^{-\mu K h}\left(D_{\mathrm{KL}}(\rho_0 \mid \pi) + \frac{1 - e^{-\mu h}}{\mu} C\mathbf{G}_0\right) + \frac{1 - e^{-K\mu h}}{\mu}\tau, \tag{77}$$

this proves Theorem 4.1

### C.5 Complexity of Langevin-MARINA

If $h \leq \min\{\frac{1}{14L}\sqrt{\frac{p}{1+\alpha}}, \frac{p}{6\mu}\}$, we wil have $C = \mathcal{O}(\frac{1}{p})$. To achieve $D_{\mathrm{KL}}(\rho_K \mid \pi) \leq e^{-\mu K h}\Psi_3 + \frac{1 - e^{-\mu K h}}{\mu}\tau = \mathcal{O}(\varepsilon)$, we need to bound the residual term $\frac{1 - e^{-K\mu h}}{\mu}\tau$ and the contraction term $e^{-\mu K h}\Psi_3$ respectively. When $\theta = 0$,

$h = \mathcal{O}\left(\frac{1}{\frac{4L^2(1+\alpha)d}{\mu p\varepsilon} + \sqrt{\frac{8L^3(1+\alpha)d}{\mu p\varepsilon}}}\right) = \mathcal{O}\left(\frac{\mu p\varepsilon}{L^2(1+\alpha)d}\right)$ is enough to make $\frac{1 - e^{-K\mu h}}{\mu}\tau = \mathcal{O}(\varepsilon)$; when $\theta \neq 0$, we need

further require $b = \Omega(\frac{\sum_{i=1}^n \sigma_i^2}{\mu n^2 \varepsilon})$ to make the left term $\frac{1 - e^{-K\mu h}}{\mu} C\theta = \mathcal{O}(\varepsilon)$. For the contraction term, we need

$K = \Omega\left(\frac{L^2(1+\alpha)d}{\mu^2 p\varepsilon}\log(\frac{\Psi_3}{\varepsilon})\right)$ (here we assume $\frac{\mu p\varepsilon}{L^2(1+\alpha)d} \ll \min\left\{\frac{1}{14L}\sqrt{\frac{p}{1+\alpha}}, \frac{p}{6\mu}\right\}$) to make $e^{-\mu K h}\Psi_3 = \mathcal{O}(\varepsilon)$.

So for all gradient estimators, we need $K = \Omega\left(\frac{L^2(1+\alpha)d}{\mu^2 p\varepsilon}\log(\frac{\Psi_3}{\varepsilon})\right)$ and $h = \mathcal{O}\left(\frac{\mu p\varepsilon}{L^2(1+\alpha)d}\right)$ to guarantee

$D_{\mathrm{KL}}(\rho_K \mid \pi) = \mathcal{O}(\varepsilon)$, for algorithm based on equation 17, we need further assume $b = \Omega(\frac{\sum_{i=1}^n \sigma_i^2}{\mu n^2 \varepsilon})$.

The analysis of the complexity under the Total Variation distance and the 2-Wasserstein distance are similar. We want $\|\rho_K - \pi\|_{TV}^2 \leq \mathcal{O}(\varepsilon^2)$ and $W_2^2(\rho_K, \pi) \leq \mathcal{O}(\varepsilon^2)$, by Theorem 4.2 we only need to guarantee $e^{-\mu K h}\Psi_3 + \frac{1 - e^{-\mu K h}}{\mu}\tau = \mathcal{O}(\varepsilon^2)$ and $e^{-\mu K h}\Psi_3 + \frac{1 - e^{-\mu K h}}{\mu}\tau = \mathcal{O}(\mu\varepsilon^2)$ respectively. So we only need to replace the $\varepsilon$ in $K, h, b$ in the above by $\varepsilon^2$ and $\mu\varepsilon^2$, then we will have $\|\rho_K - \pi\|_{TV} = \mathcal{O}(\varepsilon)$ and $W_2(\rho_K, \pi) = \mathcal{O}(\varepsilon)$.

### C.6 Proofs of Theorem B.1 and B.2

Define two flows,

$$x_t = x_k - t g_k, \tag{78}$$

$$x_t^s = \begin{cases} x_t & s < t \\ x_t - \int_t^s \nabla F(x_t^l)dl & s \geq t \end{cases}, \tag{79}$$

where $g_k$ is the gradient estimator of equation 14, equation 16 or equation 17, $x_0 = x_k, x_h = x_{k+1}$, note $x_t^s$ is continuous with respect to $s$ and when $s = t$, $x_t^s = x_t$.

Then follow the same procedure as in the last section, we have

$$
\begin{aligned}
\frac{dF(x_t)}{dt} &= \frac{dF(x_t^s)}{ds}\big|_{s=t} + \left(\frac{dF(x_t)}{dt} - \frac{dF(x_t^s)}{ds}\big|_{s=t}\right) \\
&= -\|\nabla F(x_t)\|^2 + \langle \nabla F(x_t), \nabla F(x_t) - g_k \rangle \\
&\leq -\|\nabla F(x_t)\|^2 + \frac{1}{4}\|\nabla F(x_t)\|^2 + \|\nabla F(x_t) - g_k\|^2 \\
&\leq -\frac{3}{4}\|\nabla F(x_t)\|^2 + \|\nabla F(x_t) - g_k\|^2 \\
&\leq -\frac{3}{4}\|\nabla F(x_t)\|^2 + 2\|\nabla F(x_t) - \nabla F(x_k)\|^2 + 2\|\nabla F(x_k) - g_k\|^2 \\
&\leq -\frac{3}{4}\|\nabla F(x_t)\|^2 + 2L^2\|x_{k+1} - x_k\|^2 + 2\|\nabla F(x_k) - g_k\|^2,
\end{aligned}
\tag{80}
$$

since $\|\nabla F(x_t) - \nabla F(x_k)\|^2 \leq L^2 t^2 \|g_k\|^2 \leq L^2 h^2 \|g_k\|^2 = L^2 \|x_{k+1} - x_k\|$. We define $\mathbf{G}_k := \mathrm{E}\left[\|\nabla F(x_k) - g_k\|^2\right]$, then take expectation on both sides of equation 80, we have

$$
\frac{d\mathrm{E}\left[F(x_t)\right]}{dt} \leq -\frac{3}{4}\mathrm{E}\left[\|\nabla F(x_t)\|^2\right] + 2L^2\mathrm{E}\left[\|x_{k+1} - x_k\|^2\right] + 2\mathbf{G}_k.
\tag{81}
$$

Next, we bound $\mathrm{E}\left[\|x_{k+1} - x_k\|^2\right]$,

$$
\begin{aligned}
\mathrm{E}\left[\|x_{k+1} - x_k\|^2\right] &= h^2\mathrm{E}\left[\|g_k\|^2\right] \\
&\leq 2h^2\mathrm{E}\left[\|\nabla F(x_k)\|^2\right] + 2h^2\mathrm{E}\left[\|\nabla F(x_k) - g_k\|^2\right] \\
&\leq 4h^2\mathrm{E}\left[\|\nabla F(x_t)\|^2\right] + 4h^2\mathrm{E}\left[\|\nabla F(x_t) - \nabla F(x_k)\|^2\right] + 2h^2\mathbf{G}_k \\
&\leq 4h^2\mathrm{E}\left[\|\nabla F(x_t)\|^2\right] + 4L^2 h^2\mathrm{E}\left[\|x_{k+1} - x_k\|^2\right] + 2h^2\mathbf{G}_k,
\end{aligned}
\tag{82}
$$

which is equivalent to

$$
\left(1 - 4L^2 h^2\right)\mathrm{E}\left[\|x_{k+1} - x_k\|^2\right] \leq 4h^2\mathrm{E}\left[\|\nabla F(x_t)\|^2\right] + 2h^2\mathbf{G}_k.
\tag{83}
$$

If we let $h \leq \frac{1}{2\sqrt{2}L}$, then based on equation 83, we have

$$
\mathrm{E}\left[\|x_{k+1} - x_k\|^2\right] \leq 8h^2\mathrm{E}\left[\|\nabla F(x_t)\|^2\right] + 4h^2\mathbf{G}_k.
\tag{84}
$$

Add $C\mathbf{G}_{k+1}$ to both sides of equation 81, $C$ is some parameter to be determined later. Then we calculate the right hand side based on Theorem 3.1,

$$
\begin{aligned}
RHS &= -\frac{3}{4}\mathrm{E}\left[\|\nabla F(x_t)\|^2\right] + 2L^2\mathrm{E}\left[\|x_{k+1} - x_k\|^2\right] + 2\mathbf{G}_k + C\mathbf{G}_{k+1} \\
&\leq -\frac{3}{4}\mathrm{E}\left[\|\nabla F(x_t)\|^2\right] + 2L^2\mathrm{E}\left[\|x_{k+1} - x_k\|^2\right] + 2\mathbf{G}_k + C\left((1-p)\mathbf{G}_k + (1-p)L^2\alpha\mathrm{E}\left[\|x_{k+1} - x_k\|^2\right] + \theta\right) \\
&\leq -\frac{3}{4}\mathrm{E}\left[\|\nabla F(x_t)\|^2\right] + \left(2L^2 + C(1-p)L^2\alpha\right)\mathrm{E}\left[\|x_{k+1} - x_k\|^2\right] + \left(2 + C(1-p)\right)\mathbf{G}_k + C\theta \\
&\leq -\frac{3}{4}\mathrm{E}\left[\|\nabla F(x_t)\|^2\right] + \left(2L^2 + C(1-p)L^2\alpha\right)\left(8h^2\mathrm{E}\left[\|\nabla F(x_t)\|^2\right] + 4h^2\mathbf{G}_k\right) + \left(2 + C(1-p)\right)\mathbf{G}_k + C\theta \\
&\leq -\left(\frac{3}{4} - 8h^2\left(2L^2 + C(1-p)L^2\alpha\right)\right)\mathrm{E}\left[\|\nabla F(x_t)\|^2\right] + \left(8L^2 h^2 + C(1-p)\left(4L^2 h^2\alpha + 1\right) + 2\right)\mathbf{G}_k + C\theta.
\end{aligned}
\tag{85}
$$

Choose $\beta$ (we will set $\beta = 1$ or $\beta = e^{\mu h}$ in the later)  and let $C = \left(8L^2h^2 + C(1-p)\left(4L^2h^2\alpha + 1\right) + 2\right)\beta$, solve this equation, we get

$$C = \frac{8L^2h^2\beta + 2\beta}{1 - (1-p)\left(4L^2h^2\alpha + 1\right)\beta}. \tag{86}$$

We need

$$C \geq 0, \quad \frac{3}{4} - 8h^2\left(2L^2 + C(1-p)L^2\alpha\right) \geq \frac{1}{2}. \tag{87}$$

By similar analysis as in Appendix C.4, we need require $h \leq \frac{1}{10L}\sqrt{\frac{p}{1+\alpha}}$ when $\beta = 1$ and $h \leq \min\{\frac{1}{14L}\sqrt{\frac{p}{1+\alpha}}, \frac{p}{6\mu}\}$ when $\beta = e^{\mu h}$ to guarantee equation 87. Once we have equation 87, then we finally have by equation 85

$$\frac{d\mathrm{E}\left[F(x_t)\right]}{dt} \leq -\frac{1}{2}\mathrm{E}\left[\|\nabla F(x_t)\|^2\right] + \beta^{-1}CG_k - C\mathbf{G}_{k+1} + C\theta. \tag{88}$$

**Case I.** Let $\beta = 1$, then we have

$$\frac{d\mathrm{E}\left[F(x_t)\right]}{dt} + C\mathbf{G}_{k+1} \leq -\frac{1}{2}\mathrm{E}\left[\|\nabla F(x_t)\|^2\right] + C\mathbf{G}_k + C\theta, \tag{89}$$

where $C$ is defined in equation 86, integrating both sides from $0$ to $h$, we have

$$\mathrm{E}\left[F(x_{k+1})\right] - \mathrm{E}\left[F(x_k)\right] + hC\mathbf{G}_{k+1} \leq -\frac{1}{2}\int_0^h \mathrm{E}\left[\|\nabla F(x_t)\|^2\right]dt + hC\mathbf{G}_k + hC\theta, \tag{90}$$

which is equivalent to

$$\int_0^h \mathrm{E}\left[\|\nabla F(x_t)\|^2\right]dt \leq 2\left(\left(\mathrm{E}\left[F(x_k)\right] + hC\mathbf{G}_k\right) - \left(\mathrm{E}\left[F(x_{k+1})\right] + hC\mathbf{G}_{k+1}\right)\right) + 2hC\theta. \tag{91}$$

If we first do the same procedure as above for $k = 0, 1, 2, \cdots, K-1$ then take summation from both sides, we will have

$$\begin{aligned}
\frac{1}{Kh}\int_0^{Kh} \mathrm{E}\left[\|\nabla F(x_t)\|^2\right]dt &\leq \frac{2\left(\left(\mathrm{E}\left[F(x_0)\right] + hC\mathbf{G}_0\right) - \left(\mathrm{E}\left[F(x_K)\right] + hC\mathbf{G}_K\right)\right)}{Kh} + 2C\theta \\
&\leq \frac{2\left(F(x_0) + hC\mathbf{G}_0 - F(x^*)\right)}{Kh} + 2C\theta,
\end{aligned} \tag{92}$$

this proves Theorem B.1.

**Case II.** Suppose Lojasiewicz condition holds, that is

$$\|\nabla F(x)\|^2 \geq 2\mu\left(F(x) - \min F\right), \quad \forall x \in \mathbb{R}^d. \tag{93}$$

Combine equation 88 and equation 93, we have

$$\frac{d\left(\mathrm{E}\left[F(x_t)\right] - F(x^*)\right)}{dt} \leq -\mu\left(\mathrm{E}\left[F(x_t)\right] - F(x^*)\right) + \beta^{-1}CG_k - C\mathbf{G}_{k+1} + C\theta, \tag{94}$$

which is equivalent to the integral form

$$\mathrm{E}\left[F(x_t)\right] - F(x^*) \leq \left(\beta^{-1}CG_k - C\mathbf{G}_{k+1} + C\theta\right)t + \mathrm{E}\left[F(x_k)\right] - F(x^*) + \int_0^t -\mu\left(\mathrm{E}\left[F(x_\tau)\right] - F(x^*)\right)d\tau, \quad t \in [0, h]. \tag{95}$$

Now we use Grönwall inequality C.1, note equation 95 satisfies equation 46 with $\phi(t) = F(x_t) - F(x^*)$, $B(t) = \left(\beta^{-1}CG_k - C\mathbf{G}_{k+1} + C\theta\right)t + F(x_k) - F(x^*)$, $C(t) = -\mu$, then by equation 47, we have

$$\mathrm{E}\left[F(x_t)\right] - F(x^*) \leq e^{-\mu t}\left(\mathrm{E}\left[F(x_k)\right] - F(x^*)\right) + \frac{1 - e^{-\mu t}}{\mu}\left(\beta^{-1}CG_k - C\mathbf{G}_{k+1} + C\theta\right), \tag{96}$$

let $t = h$ and $\beta = e^{\mu h}$, then we have

$$\mathrm{E}\left[F(x_{k+1})\right] - F(x^*) + \frac{1-e^{-\mu h}}{\mu}C\mathbf{G}_{k+1} \leq e^{-\mu h}\left(\mathrm{E}\left[F(x_k)\right] - F(x^*) + e^{\mu h}\frac{1-e^{-\mu h}}{\mu}\beta^{-1}C\mathbf{G}_k\right) + \frac{1-e^{-\mu h}}{\mu}C\theta$$

$$= e^{-\mu h}\left(\mathrm{E}\left[F(x_k)\right] - F(x^*) + \frac{1-e^{-\mu h}}{\mu}C\mathbf{G}_k\right) + \frac{1-e^{-\mu h}}{\mu}C\theta,$$

(97)

use equation 97 for $k = 0, 1, 2, \cdots, K-1$, we have finally

$$\mathrm{E}\left[F(x_K)\right] - F(x^*) \leq \mathrm{E}\left[F(x_K)\right] - F(x^*) + \frac{1-e^{-\mu h}}{\mu}C\mathbf{G}_K$$

$$\leq e^{-\mu K h}\left(\mathrm{E}\left[F(x_0)\right] - F(x^*) + \frac{1-e^{-\mu h}}{\mu}C\mathbf{G}_0\right) + \frac{1-e^{-K\mu h}}{\mu}C\theta \qquad (98)$$

$$\leq e^{-\mu K h}\left(F(x_0) + \frac{1-e^{-\mu h}}{\mu}C\mathbf{G}_0 - F(x^*)\right) + \frac{1-e^{-K\mu h}}{\mu}C\theta,$$

this proves Theorem B.2.

