# OpenReview forum: "Federated Sampling with Langevin Algorithm under Isoperimetry"
_TMLR — Accepted by TMLR_

### Review · Reviewer_M5MN · 2023-08-29

**Summary Of Contributions:**

The paper proposes a Langevin algorithm for federated learning, along with its convergence analysis under log Sobolev inequality.

**Audience:**

Yes

**Broader Impact Concerns:**

No concern.

**Claims And Evidence:**

Yes

**Requested Changes:**

The paper would benefit from a thorough comparison with more commonly used federated learning algorithms, both in theory and in experiments.

**Strengths And Weaknesses:**

The strength is that analyses are given to the proposed algorithm, showing improvement over the previous works on communication cost.
Weaknesses are mainly in three folds.
1. The assumed log Sobolev inequality is a pretty strong assumption due to its strong connection to gradient dominance and strongly concave conditions.
2. There is no comparison with commonly used FedAvg style algorithms in federated learning, both in terms of communication and convergence guarantee.
3. No experiments in the paper to validate the actual performance of the proposed algorithm.

---

> ### Author Response · Authors · 2023-10-11
> **Answers**
>
> Thanks for the review. We address your three points below.
>
> 1. **The assumed log Sobolev inequality is a pretty strong assumption due to its strong connection to gradient dominance and strongly concave conditions.**
>
> We do not think that log Sobolev is a strong condition because:
>
> - There is no mathematical relationship between Log Sobolev and gradient dominance as the reviewer says. The connection between the two is used for *intuition* only to guide us in the proofs.
>
> - Log Sobolev is milder than strong convexity. Strong convexity implies log Sobolev, but the converse is false, see the paragraph after Assumption 2.1.
>
> In summary, log Sobolev is milder than other conditions used in the literature: existing literature on Federated Sampling relies on strong convexity (see Table 1)
>
>
> 2. **There is no comparison with commonly used FedAvg style algorithms in federated learning, both in terms of communication and convergence guarantee.**
>
> We have a whole section dedicated to this question, see Section 4.2. Comparison to FedAvg is not relevant here because FedAvg is an optimization algorithm, whereas MARINA-Langevin is a sampling algorithm.
>
> 3. **No experiments in the paper to validate the actual performance of the proposed algorithm.**
>
> Preliminary experiments are located in the appendix, Section A.

---

### Review · Reviewer_ZmLB · 2023-08-29

**Summary Of Contributions:**

This paper presents a federated algorithm for sampling distributions of the form $\\mu \\propto \\exp(-F)$, assuming that they satisfy a log-sobolev inequality. The paper presents an analysis for this algorithm, proving that the procedure samples from a distribution that is close in KL divergence to $\\mu$.

The analysis proceeds by adapting the analysis of MARINA (which was an algorithm previously proposed by Gorbunov et al. (2021) for federated optimization) to the setting of sampling, by adapting a proof of Vempala & Wibisono (2019) for the convergence of Langevin sampling.

One of the main aspects that differentiates this work from previous works on federated sampling is that this work applies to distributions satisfying an LSI, while previous work along these lines assumed the more restrict condition of strong convexity.

**Audience:**

Yes

**Claims And Evidence:**

Yes

**Requested Changes:**

As well as what I wrote above in the weaknesses section, please address these typos, and questions:

* Typo: "Proof of Theorem 3.3" and "Theorem 3.4" in the appendix, when these are Proposition 3.3 and Proposition 3.4 in the main text

* Comment: In Section 3.3.1, make it more clear that all nodes coordinate in either uploading $\\nabla F_i(x_{k+1})$ or $g_k + \\mathcal{Q}(\\nabla F_i(x_{k+1}) - \\nabla F_i(x_k))$. I.e., the choices are not independent. (unless I am misreading the proof here, this is necessary)

* Typo: some equalities in proof of Prop. 3.4 should be inequalities

* Comment: On page 7, please make it clear that $\\mathcal{I}'_{i,k}$ is a multiset with elements chosen with replacement from $[N]$.

* Typo: Equation (39) has a typo with $\\rangle$ vs. $|$ on the 5th line.

* Typo: Equation (39), $\\chi_s'$ should be $\\chi_{s'}$

* Question: "Therefore the communication complexity is higher..." on page 8. This is only higher by a factor of 2, no?

* Typo: Equation (21), $H_{\\pi}(\\sigma)$ is not defined. You meant $D_{KL}$ instead?

* Typo: In (59), $J_{\\pi}$ should be $J_{\\pi}(\rho_t)$

**Strengths And Weaknesses:**

Strengths.
The paper is on the whole clearly written and well structured. Furthermore, the math appears correct. The result should be of interest to the sampling community, and is therefore pertinent to TMLR.

Weaknesses.
I am inclined to recommend acceptance of the paper as it is, but there is one addition that I would like the authors to make. After Corollary 4.2, the authors discuss the communication cost of their protocol and compare it to the communication cost of previous works. However, I found this discussion difficult to read and the comparison to previous works difficult to interpret because certain quantities like $p$ are not made explicit, and are left as a choice to the reader.

I would like the authors to add calculations of the communication cost in the specific case that the compression operator $\\mathcal{Q}$ is the operator that rounds each number to $B$ bits. What are the optimal choices of $B$ and $p$ in that case to minimize the communication? And what is the total communication complexity cost in terms of # of bits? And how does that compare to previous works? This would help me (and future readers) immensely to understand the significance of the contribution. (Alternatively, another compression operator $\\mathcal{Q}$ could also be analyzed, but the point is that there should be a specific example of the implication of the theorem.)

---

> ### Author Response · Authors · 2023-10-11
> **Communication cost**
>
> Thanks for your thorough review, and for the typos/questions. We will correct them in the revised version of the paper.
>
> **I would like the authors to add calculations of the communication cost in the specific case that the compression operator is the operator that rounds each number to B bits**
>
> Thanks for this suggestion. It is true that, because we work at the abstract level of MARINA estimators, comparison to concurrent works in terms of the communication complexity is not straightforward. We are adding the requested calculation in our revision along with comparison with the literature. We believe this is a great addition to the paper.

---

### Review · Reviewer_P7nJ · 2023-09-24

**Summary Of Contributions:**

This paper studies federated sampling problems. In particular, it extended the standard MARINA algorithm to the sampling problem and provided the convergence rate.

However, the current submission does not demonstrate the unique challenges when extending MARINA to Langevin-MARINA, and how these unique challenges are addressed. From the current version, it seems the extension is not straightforward. It would be good if the authors could provide more discussions about this critical point.

**Audience:**

Yes

**Claims And Evidence:**

No

**Requested Changes:**

The contributions of this submission are not clear.
1. To improve the convergence rate of the standard MARINA algorithm, what is the unique challenge here? What new techniques are developed to address these challenges?

2. When extending MARINA to Langevin-MARINA, what are the unique challenges here? What new techniques are developed to address these challenges?

3. Compared with existing federated sampling approaches, what are the unique challenges when considering the weaker log Sobolev inequality?

**Strengths And Weaknesses:**

Strengths:
1. It improved the convergence rate of the standard MARINA algorithm.

2. It investigates a new setting: the weaker log Sobolev inequality.

Weakness:
The contributions of this submission are not clear.
1. To improve the convergence rate of the standard MARINA algorithm, what is the unique challenge here? What new techniques are developed to address these challenges?

2. When extending MARINA to Langevin-MARINA, what are the unique challenges here? What new techniques are developed to address these challenges?

3. Compared with existing federated sampling approaches, what are the unique challenges when considering the weaker log Sobolev inequality?

---

> ### Author Response · Authors · 2023-10-11
> **Challenges**
>
> Thanks for your positive review. We address your questions here and will update our paper accordingly.
>
> **To improve the convergence rate of the standard MARINA algorithm, what is the unique challenge here? What new techniques are developed to address these challenges?**
>
> Our proofs regarding the standard MARINA algorithm employs a dynamical point of view. We consider a path $(x_t)$ that goes from $x_k$ to $x_{k+1}$ and we bound the derivative of $F(x_t)$ along this path. Later we integrate along this path to bound the objective gap.
>
> Therefore, the challenges are to (i) consider a dynamical proof (ii) "run" this dynamical proof. For (i), the idea of this proof was inspired to us by the analysis of Langevin MARINA for which differentiating along a path seems necessary. For (ii), we used inequalities similar to the ones used in the analysis of MARINA, but in a different way.
>
>
> **When extending MARINA to Langevin-MARINA, what are the unique challenges here? What new techniques are developed to address these challenges?**
>
> It is difficult to explain the challenge without being too technical. Basically, the main challenge is to be able to develop a Langevin-MARINA algorithm that exhibits both the advantages of Langevin and MARINA without losing anything. We addressed this challenge by bounding the RHS of Eq (56) (see page 24) in a novel manner, using the MARINA estimator. This RHS would have been bounded differently for the vanilla Langevin algorithm.
>
> **Compared with existing federated sampling approaches, what are the unique challenges when considering the weaker log Sobolev inequality?**
>
> The main challenge is the lack of convexity of $F$. To our knowledge, existing federated sampling rely of $F$ convex, whereas we do not. Because of this lack of convexity we cannot hope to bound the distance between the iterate and the solution in Wasserstein distance directly. Therefore, we make a detour to first bound the KL, which plays the role of an objective gap.

---

### Decision · Action_Editor_FSyY · 2024-01-07

**Recommendation:** Accept as is

**Comment:**

The submission meets both *claim and evidence* and *audience* criteria. However, the reviewers were generally lukewarm due to the incremental contribution and lack of experiments [see more above]. Only one reviewer engaged in the discussion/rebuttal phase, 2/3 reviewers proposed *leaning accept*.

**Audience:**

Yes.

**Claims And Evidence:**

The submission proposed a Langevin variant of MARINA, extending MARINA's scope from optimisation to posterior sampling. Overall, the reviewers were happy with the proposed algorithm (most agreed that it was incremental, however) and the theoretical convergence analysis.

Only one synthetic bimodal 1D illustration was provided in the appendix so the reviewers were not convinced about the practical efficacy in more complex, realistic federated learning settings.